# Discrimination of pollen of New Zealand mānuka (*Leptospermum scoparium* agg.) and kānuka (*Kunzea* spp.) (Myrtaceae)

X. Li[1]*, J. G. Prebble[1], P. J. de Lange[2], J. I. Raine[1], L. Newstrom-Lloyd[3]

**1** GNS Science, Lower Hutt, New Zealand, **2** School of Environmental & Animal Sciences, Unitec Institute of Technology, Auckland, New Zealand, **3** The New Zealand Trees for Bees Research Trust, Havelock North, New Zealand

* x.li@gns.cri.nz

## Abstract

The very similar appearance of pollen of the New Zealand Myrtaceous taxa *Leptospermum scoparium* s.l. (mānuka) and *Kunzea* spp. (kānuka) has led palynologists to combine them in paleoecological and melissopalynological studies. This is unfortunate, as differentiation of these taxa would improve understanding of past ecological change and has potential to add value to the New Zealand honey industry, where mānuka honey attracts a premium price. Here, we examine in detail the pollen morphology of the 10 *Kunzea* species and a number of *Leptospermum scoparium* morphotypes collected from around New Zealand, using light microscopy, SEM, and Classifynder (an automated palynology system). Our results suggest that at a generic level the New Zealand *Leptospermum* and *Kunzea* pollen can be readily differentiated, but the differences between pollen from the morphotypes of *Leptospermum* or between the species of *Kunzea* are less discernible. While size is a determinant factor–equatorial diameter of *Leptospermum scoparium* pollen is 19.08 ± 1.28 μm, compared to 16.30 ± 0.95 μm for *Kunzea* spp.–other criteria such as surface texture and shape characteristics are also diagnostic. A support vector machine set up to differentiate *Leptospermum* from *Kunzea* pollen using images captured by the Classifynder system had a prediction accuracy of ~95%. This study is a step towards future melissopalynological differentiation of mānuka honey using automated pollen image capture and classification approaches.

## Introduction

Honey from New Zealand mānuka (Myrtaceae: *Leptospermum scoparium* J.R.Forst. et G.Forst. agg.) attracts a premium value [1–3], arising from the medical benefits of its non-peroxide antibacterial activity [4]. This has created a demand from industry and regulators for accurate and cost-effective authentication testing of the product [5].

   Melissopalynology–the pollen analysis of honey–is widely applied internationally, especially in Europe and America, in combination with chemical, physical, and sensory characters to indicate the approximate contributions of nectar from various plants to the honey [6–9]. However, using melissopalynology techniques to determine the nectar contribution of

**Funding:** An initial study of pollen of male and hermaphrodite flowers by JIR was funded by New Zealand Ministry for Primary Industry. XL received funding [no serial number] from KiwiNet (https://www.mbie.govt.nz/science-and-technology/science-and-innovation/funding-information-and-opportunities/investment-funds/preseed-accelerator-fund/kiwi-innovation-network-limited/) and GNS Science SSIF. The funders had no role in study design, data collection and analysis, decision to publish, or preparation of the manuscript.

**Competing interests:** The authors have declared that no competing interests exist.

*Leptospermum scoparium* in New Zealand honeys has been hampered by the very similar appearance of *Leptospermum scoparium* pollen to pollen from New Zealand *kānuka* (*Kunzea* spp.). *Kunzea* Rchb., the closest relative to *Leptospermum* J.R.Forst. et G.Forst. in New Zealand [10], is widely distributed throughout the two main islands of New Zealand [11] and is often found growing with *Leptospermum* J.R.Forst. et G. Forst. It is also frequently visited by honey-bees (*Apis mellifera* Linnaeus, 1758).

Until the 1980s three species of *Leptospermum* were regarded as endemic to New Zealand, *L. scoparium* J.R.Forst. et G.Forst. ("mānuka"), *L. ericoides* A.Rich. ("kānuka") and *L. sinclairii* Kirk [12]. Thompson transferred *L. ericoides* to *Kunzea*, treating *L. sinclairii* as a synonym [13]. After a full taxonomic revision of the *Kunzea ericoides* complex including morphological, cytological and molecular variation as well as hybridisation experiments, de Lange [11] recognised ten *Kunzea* species, all endemic to New Zealand. Variation in *Leptospermum scoparium* is also well known, for example in flower colour, growth habit and yield of the antibacterial precursor chemical DHA (dihydroxyacetone) in nectar [12, 14–17]. Nevertheless, until recently only one species *L. scoparium* with two varieties, *L. scoparium* var. *incanum* Cockayne and *L. scoparium* var. *scoparium* have been accepted for New Zealand [12, 16, 18]. In her Australasian revision of the genus Thompson [13] only accepted the one species, *L. scoparium* for New Zealand but she did not critically examine New Zealand plants. At the time of writing, a New Zealand Department of Conservation funded revision of that nation's *Leptospermum scoparium* is underway. Initial results confirm the findings of others, notably Buys et al. [19] that New Zealand populations of *L. scoparium* are distinct from those populations in Australia and Tasmania attributed to *L. scoparium*, and so are endemic. Buys et al. [19] also suggested that New Zealand plants are worthy of taxonomic segregation (see [16] and references therein). While the findings from the taxonomic revision of New Zealand *L. scoparium* are still pending, the morphotype analysed in this paper as *L.* aff. *scoparium* (c) "Waikato Peat Bog" has already been established as the new species *L. repo* de Lange et L.M.H.Schmid, and others are pending [16].

Because of their wide geographical distribution, and wide ecological envelope, pollen records of *Leptospermum* and *Kunzea* have received little focus in New Zealand palaeovegetation studies. As a result, limited effort has been assigned to pollen morphological study of the two forms. For example, in his seminal work on melissopalynology of New Zealand honeys Moar [20] referred to both forms simply as *Leptospermum* (although this was prior to Thompson's 1989 transfer of *L. ericoides* to *Kunzea*), and in vegetation history studies he later classed them together as *Leptospermum* type [21]. However, differences in pollen morphology (at least on a statistical basis) had previously been observed by Pike [22], McIntyre [23], and Harris et al. [24], and later by Moar himself [25].

The increasing economic importance of mānuka honey has led to new focus on *Leptospermum* and *Kunzea* pollen [2]. In 2014, the Ministry for Primary Industries initiated preliminary studies to set up a standard for manuka honey. As part of this, we carried out a pilot study to investigate the possibility of differences between pollen of male and hermaphrodite (bisexual) flowers of New Zealand *Leptospermum* and *Kunzea*, both of which are at least in part andromonoecious [11, 26] (pers. obs. Newstrom-Lloyd). It was thought possible that the pollen of male and bisexual flowers may differ slightly in morphology, for instance in size, as has been noted in other andromonoecious plants (e.g., *Sagittaria guayanensis* Kunth [27]). Based on a small number of plant specimens of *Kunzea robusta* de Lange et Toelken and *Leptospermum scoparium* s.l. from the East Cape region we found no discernible difference in pollen of the two flower types within each genus, but a significant difference in average size between pollen of the two genera [28]. This conclusion agreed with the previous results of McIntyre [23] and Harris et al. [24] and was further confirmed by measurements of pollen from a small number of plant samples by Holt & Bebbington [29]. The results of our pilot study are summarised in S1 File.

As a result of these findings, in 2016 we set out to examine in detail the pollen morphology of the 10 New Zealand *Kunzea* species recently established by de Lange [11], and pollen from *Leptospermum scoparium* and putative segregates from that species, covering a wide geographic spread of populations throughout the New Zealand range, to determine:

1. if *Leptospermum scoparium* s.l. and *Kunzea* pollen can be differentiated, and

2. if subpopulations of *L. scoparium* pollen or interspecific variation in *Kunzea* exist.

A clear differentiation between New Zealand *Leptospermum* and *Kunzea* pollen could be used to establish a pollen-based mānuka honey standard and form the basis for mānuka honey melissopalynological testing. The results of our detailed study are reported here. We have not studied specimens of Australian *Leptospermum* species which occur in New Zealand as a few small, locally naturalised populations unlikely to be significant sources of honey, except possibly *L. laevigatum* on Matakana Island [30, 31].

In 2017 the New Zealand Government Ministry for Primary Industries (MPI) established an export standard for mānuka honey based on assay of characteristic chemical marker compounds and the presence of DNA of manuka pollen [1, 32], partly because of the then perceived difficulty in discrimination of *Leptospermum* and *Kunzea* pollen. At the present time, all New Zealand export honey passing the MPI tests is accepted as "mānuka" honey, without distinction as to geographic (and thus possibly specific) origin, or whether other nectar sources could be predominant. We contend that melissopalynological analysis remains a useful test because a single analysis can routinely quantify major pollen components (as well as non-pollen entities such as honey-dew fungal spores) and thereby provide a more comprehensive view of the nectar sources of a putative mānuka or kānuka honey. Pollen analysis also requires less specialised equipment and is thus more readily available to individual producers as well as overseas laboratories conversant with applying it as part of Codex Alimentarius requirements for certification of monofloral honeys. It may also be less expensive than the MPI test, especially if automated image recognition is routinely achieved.

## Materials and methods

### Sample collection

Fifteen geographic/genetic populations from *Leptospermum scoparium* were selected (see below) across New Zealand, representing 15 or fewer potential species segregates based on growth habit, leaf shape, flower size and colour, capsule shape, size and colour, and chemistry. Additionally, the 10 New Zealand species of *Kunzea* were selected, along with 10 specimens of putative interspecific *Kunzea* hybrids. In total, we sampled flowers from 135 herbarium specimens (at least 2 specimens for each *Kunzea* species and *Leptospermum scoparium* segregate were taken from both ends of their geographical range) from the Auckland War Memorial Museum herbarium (S1 and S2 Tables). The locations from which the plant specimens were collected are illustrated in Fig 1. Anthers from several fully opened flowers from each herbarium specimen were collected into pre-labelled 15 ml centrifuge tubes. Following the results of our pilot study that demonstrated no difference between the pollen from male and hermaphrodite flowers of *Leptospermum scoparium* s.l. and *Kunzea robusta* (S1 File), no attempt was made to separately select male and bisexual flowers in this study.

### *Leptospermum scoparium* putative segregates

At the time this study was initiated in 2016 a taxonomic investigation of New Zealand *Leptospermum scoparium* populations had not been initiated. However, on examination of

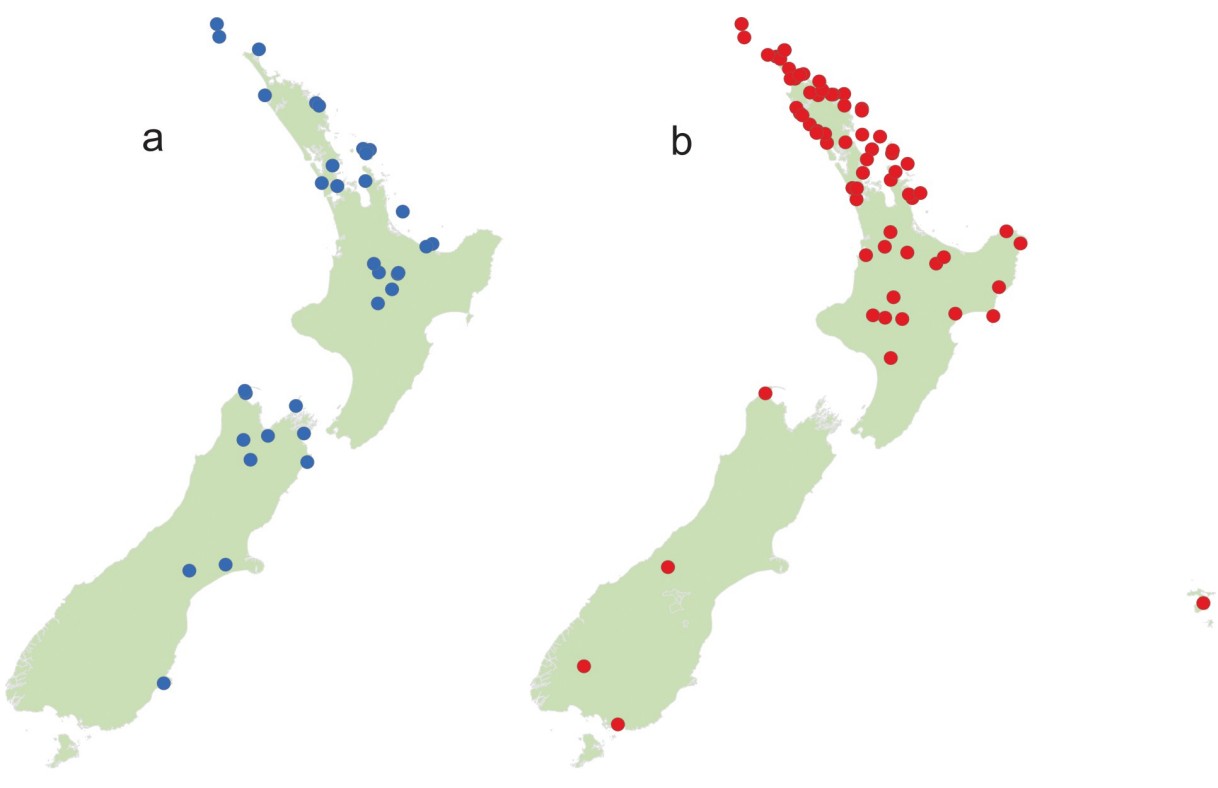

**Fig 1.** Collection localities for (a) *Kunzea* and (b) *Leptospermum scoparium* specimens.

herbarium specimens (AK) considering the morphological variation evident it was decided to split those collections into putative segregates or "morphotypes", to see if any could be differentiated by their pollen. While this investigation was in progress, four of the segregates (representing morphotypes "Auckland", "Waikato Peat Bog", "East Cape" and "Three Kings" of this paper respectively) were informally published as *L*. aff. *scoparium* (a), (c), (d), and (e) as part of a New Zealand-wide conservation assessment of the indigenous vascular flora [33]. In this paper we also examine a further 10 putative segregates ("Surville Cliffs", "Central Volcanic Plateau", "Coromandel Swamp", "Flat Silver", "Otaipango", "Papa", "South Island Mountain", "Northern South Island", "Southern South Island" and "Wellington") and examples of *L. scoparium* var. *incanum* with pink-tinged or uniformly pink flowers. Buys et al. [19] included many of these putative segregates in their molecular analysis, finding evidence to merge some of this variation into subclades within a New Zealand (endemic) *L. scoparium* clade. For the purposes of this study, whilst a taxonomic revision is in progress, we retain usage of the putative segregates as we sampled them for this study, noting that, and as mentioned above, segregate (c) is now formally recognised as *L. repo*. We also note that putative segregates recognised as "Wellington", "South Island Mountain", "Southern South Island", based on current thinking (P.J. de Lange unpubl.) may be best treated as *L. scoparium* s.s.–though this needs further study, and so we retain their usage here.

## Pollen preparation

Dried herbarium flower samples were rehydrated overnight in deionized water with several drops of filtered detergent. After being vigorously stirred, each of the sample suspensions was

sieved through a 90 μm nylon filter cloth to remove non-pollen flower parts. Pollen was then concentrated by centrifugation.

The residual pollen samples were then prepared using Erdtman's acetolysis method [34]. Each sample was washed with glacial acetic acid, heated in a 9:1 mixture of acetic anhydride and concentrated sulphuric acid for 5 minutes at 95°C in a fume hood, returned to acetic acid, then twice washed in deionized water. Microscope slides were then made using glycerine jelly with safranin stain as the mounting medium.

## Light microscopy and manual measurements

All photographs were taken with Zeiss AxioImager microscopes using 100× oil immersion objectives. Pollen grains in polar and equatorial orientations were measured either using an eyepiece graticule or using the measure function in Zen 3.2, an image-processing and analysis software from ZEISS microscopy. Visual measurements were made under the highest magnification available, using a 100× oil immersion objective, 1.6× tube factor, and a 10× eyepiece with an eyepiece graticule of 100 divisions. At this magnification, each division is equivalent to 0.625 μm. Measurements were made to the nearest half division.

Two dimensions which are usually used to characterise angiosperm pollen grain size were measured: polar diameter (Lg max, *longitudo*: Iversen and Troels-Smith 1950 [35]; P: Erdtman, Nilsson and Praglowski 1992 [34]) and equatorial diameter (Lt+, *longitudo transversa*: Iversen and Troels-Smith 1950 [35]; E: Erdtman, Nilsson and Praglowski 1992 [34]), as shown in Fig 2. Polar diameter, P, can be measured only in equatorial views of the pollen grain. Equatorial diameter, E, can be measured in suitable equatorial and oblique views, as well as in polar views. As it is constrained to be the maximum dimension along a line at right angles to a line drawn through the grain centre and the midpoint of one of the apertures, it may or may not be equivalent to the maximum Feret diameter measured by Classifynder, depending on the equatorial contour shape (see below). In the oblate pollen grains of *Leptospermum scoparium* s.l. and *Kunzea* spp., which tend to lie in polar and oblique views, it is the dimension most readily measured.

To avoid unconscious bias in the selection of pollen grains for measurement, every grain encountered in suitable orientation in one or two traverses across the middle part of the area

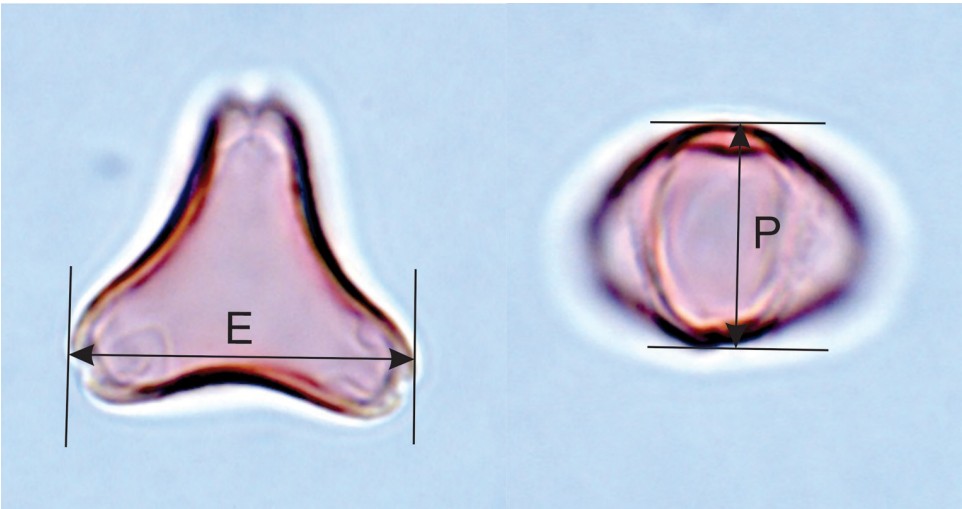

**Fig 2.** Acetolysed *Leptospermum scoparium* pollen in polar view (left) and equatorial view (right) showing equatorial (E) and polar (P) dimensions.

under the microscope slide coverslip was measured. This was continued until totals of 30 equatorial diameters and 10 polar diameters were achieved. Folded or damaged pollen grains were ignored. Specimens of pale and thin-walled pollen grains were ignored as suspected immature pollen grains.

## Scanning electron microscopy

Specimens from each of the 10 species of *Kunzea* spp. and 15 morphotypes of *Leptospermum scoparium* s.l. were selected, and all samples were prepared as described above, including dehydration, filtration and acetolysis treatment. One drop of pollen suspension from each sample was deposited on a conductive carbon tab attached to an aluminium stub and dried in a fume hood overnight. Specimens were then coated with gold, using a vacuum sputter coater, and examined using a JEOL Neoscope JCM-6000e microscope at 15 kV.

## Digital image analysis: Automated capture

The 135 pollen samples were further analysed using a Classifynder version 3a. The Classifynder is an automated palynology system [36]. It includes an automated stage and image capture system, where pollen specimens are digitally photographed under monochromatic dark field illumination using a 40× dry objective lens. Fifty parameters are calculated from each pollen image. These include 7 core parameters that are shape- and size-related: elongation, compactness, convex hull, Heywood circularity, hydraulic radius, maximum Feret diameter, and area [29]. The remaining 43 parameters are mainly texture-related but include additional measurements of shape [36, 37]. The parameters measured were selected empirically during development of the system as those best able to allow discrimination between a range of modern pollen types [37]. Here, we used the Classifynder only for image capture and measurement, but used other tools for classification, described below.

A single microscope slide was examined for each sample. The stage configuration of the Classifynder version 3a restricts analysis to a 10×10 mm square within the 20×20 mm glass cover slips used in this study. For each sample, a 10×10 mm area was scanned by the Classifynder, and digital images of all pollen-like objects were automatically collected. These false-colour images were manually sorted to extract two groups of pollen: specimens orientated in polar view, and those orientated in equatorial view. All images properly presented in these two orientations were included, including the immature pollen grains which were ignored during light microscopic manual measurements. Results reported here are based on these two subsets of images. A median number of 158 pollen grains of either equatorial or polar view was captured from 122 samples (minimum = 16, maximum = 1290 grains). Results from the other 13 samples were discarded due to either scarcity or poor presentation of pollen grains. Although 10 *Kunzea* hybrids were included in the data processing stage, the results were only used as a reference guide but not incorporated in the data analysis.

Initial inspection of parameter plots revealed a sub-population of specimens with optical parameter values far from the main population, resulting in a bimodal distribution on parameter plots. Re-examination of digital images associated with these outlying specimens revealed various imperfections, mainly poor focus. These specimens were excluded from further analysis, discarding 1190 specimens with *compactness* parameter of <0.55. This resulted in a total dataset of 17835 *Leptospermum* and 5867 *Kunzea* pollen grains, of which approximately two-thirds were images of grains in polar view. The number of observations and summary statistics are reported in S3 and S4 Tables.

Parameter data from the pollen data set was processed using packages from the open-source software R (www.r-project.org). Exploratory data analyses to investigate the extent to which

the full 50 Classifynder parameters or the 7 core parameters allowed discrimination between the various *Leptospermum scoparium* morphotypes and *Kunzea* species included discriminant function analysis (R package MASS) [38], and principal component analysis (PCA) (R package Vegan) [39]. The distribution of eigenvalues in the PCA results was illustrated using a highest posterior density region, calculated using R package *emdbook* [40].

### Digital image analysis: Machine learning

We tested the skill of a Support Vector Machine predictive model at discriminating between *Leptospermum* and *Kunzea* in our 23702 pollen grain dataset [41]. This model, which is optimised to discriminate between two classes (here two pollen types), was tested by 99 iterations of a randomised 80:20 split of the data into training and test sets. We first tested the model on the entire pollen grain dataset, using first all 50 available parameters, then the core 7 parameters. Then, we tested the model on the subset of specimens in polar view, again using all 50 available parameters, then the core 7 parameters.

## Results

The detailed pollen morphology descriptions that follow are revised from the original descriptions of McIntyre [23] and Moar [25] and are based on LM and SEM observations.

### *Leptospermum scoparium* s.l. visual observation

Pollen grains are isopolar, oblate, triangular in polar view and flattened oval in equatorial view. In polar view, the side of the amb appears to be often concave and the angles to be extended. Pollen is tricolporate and angulaperturate, very rarely di- or tetracolporate. The ectoapertures are usually narrow, and the endoapertures are narrow but lalongate. Pollen grains are normally syncolpate, but very rarely slightly parasyncolpate due to slight widening of the colpi at the poles (however, polar islands are not present). The exine is always slightly patterned: obscurely flecked under LM, but clearly scabrate under SEM (Figs 3–5). The size of the P axis is 12.98 ± 1.66 μm (n = 188) and E axis 19.08 ± 1.28 μm (n = 490).

Pollen of all 15 *Leptospermum scoparium* s.l. morphotypes were examined under LM and, except for "Otaipango", under SEM. Acknowledging there is variability within each morphotype, and even within pollen populations collected from individual plants, we make the following observations about pollen morphological characters that assist with discrimination between the morphotypes (Table 1):

1. Distinctly concave-sided pollen is more prominent in "Surville Cliffs", "Flat Silver" and "East Cape", whereas the amb of "Three Kings" and "Otaipango" pollen tends to be relatively straight.

2. Surface pattern with coarse texture, presented in patches, is especially noticeable in "Otaipango", "Northern South Island" and "Wellington", and common in "Papa", "Three Kings", *L. scoparium* var. *incanum* and "Flat Silver".

3. Vestibulum–The ectexine slightly protrudes and is buckled along the sides of the amb and forms a gap with the endexine at the apices of the amb. This feature is usually present in "Otaipango" and "Papa", and frequently seen in "Three Kings", "Surville Cliffs" and *L. scoparium* var. *incanum*.

4. Rounded and slightly enlarged apices–this feature is generally distinct in many of the morphotypes except for "Otaipango" and "Wellington".

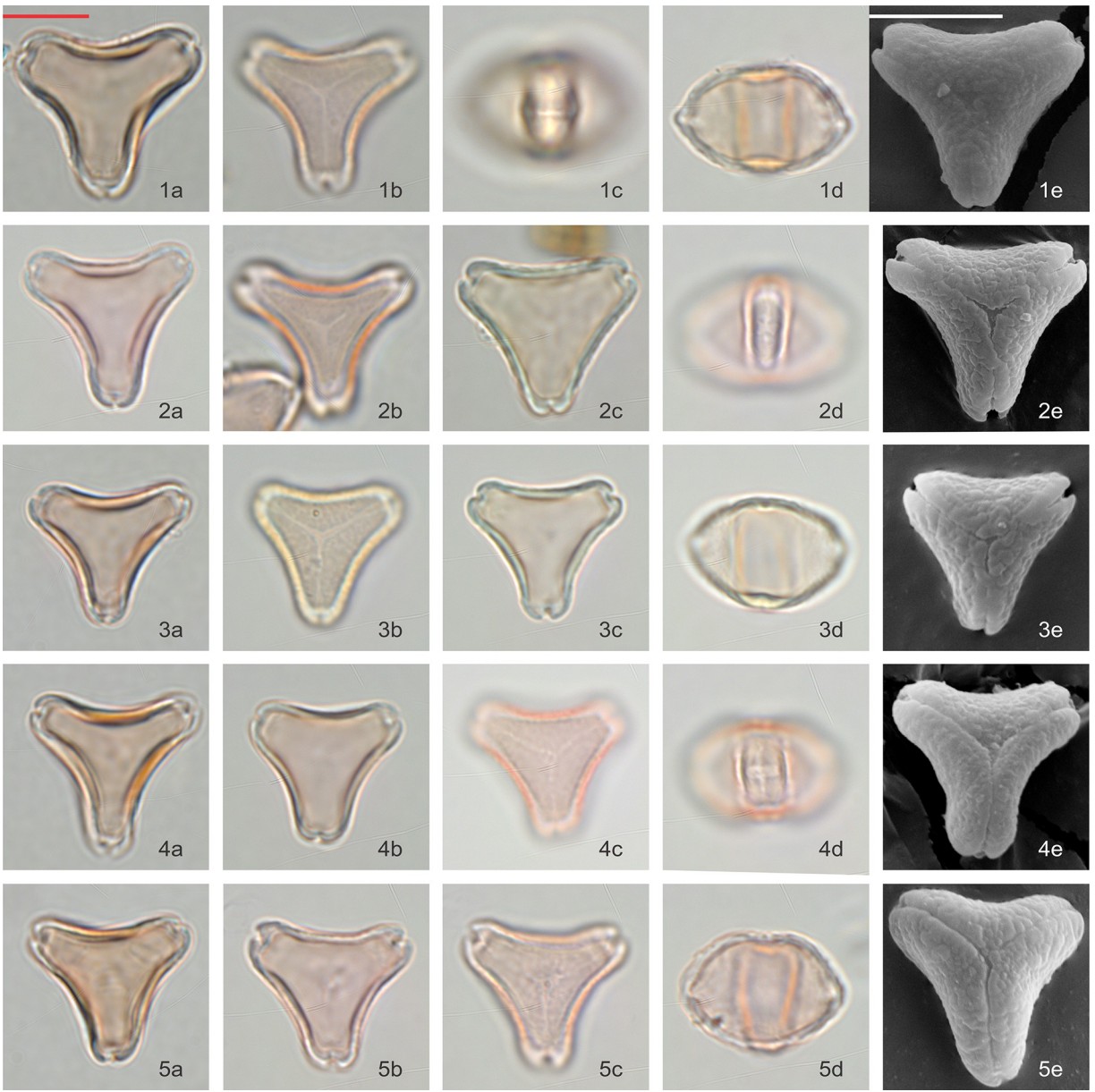

**Fig 3. Polar and equatorial views of *Leptospermum scoparium*, Part 1.** Scale bar = 10 μm. (1a-1e) Surville Cliffs, (2a-2e) Flat Silver, (3a-3e) East Cape, (4a-4e) Waikato Peat Bog, (5a-5e) Coromandel Swamp.

5. Slightly thickened arci, only resolved under SEM, are commonly found in "East Cape", "Flat Silver", "Central Volcanic Plateau", "Papa", "South Island Mountain" and "Northern South Island" populations.

6. Both "Waikato Peat Bog" and "Coromandel Swamp" have large variations especially in the shape of their pollen grains, from concave to straighten sides of amb. Tetracolporate pollen grains are frequently observed in these two morphotypes.

Among the specimens examined, "Surville Cliffs" tends to have the largest pollen with average E-axis 20.6 μm, while "Wellington" appears to have the smallest with E-axis 17. 9 μm. A

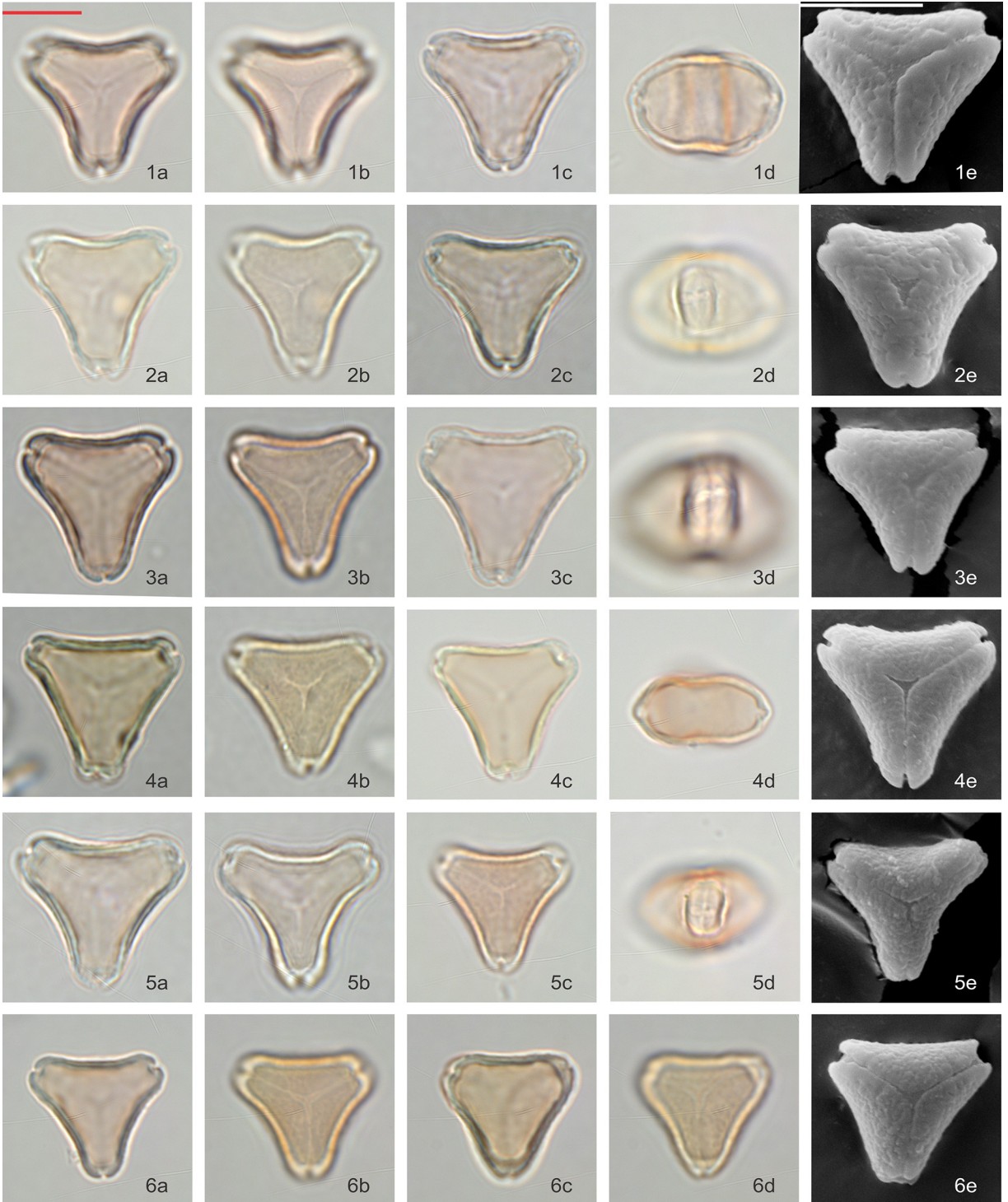

**Fig 4. Polar and equatorial views of *Leptospermum scoparium*, Part 2.** Scale bar = 10 μm. (1a-1e) South Island Mountain, (2a-2e) Central Volcanic Plateau, (3a-3e) *L. scoparium* var. *incanum*, (4a-4e) Three Kings, (5a-5e, 6a-6e) Auckland.

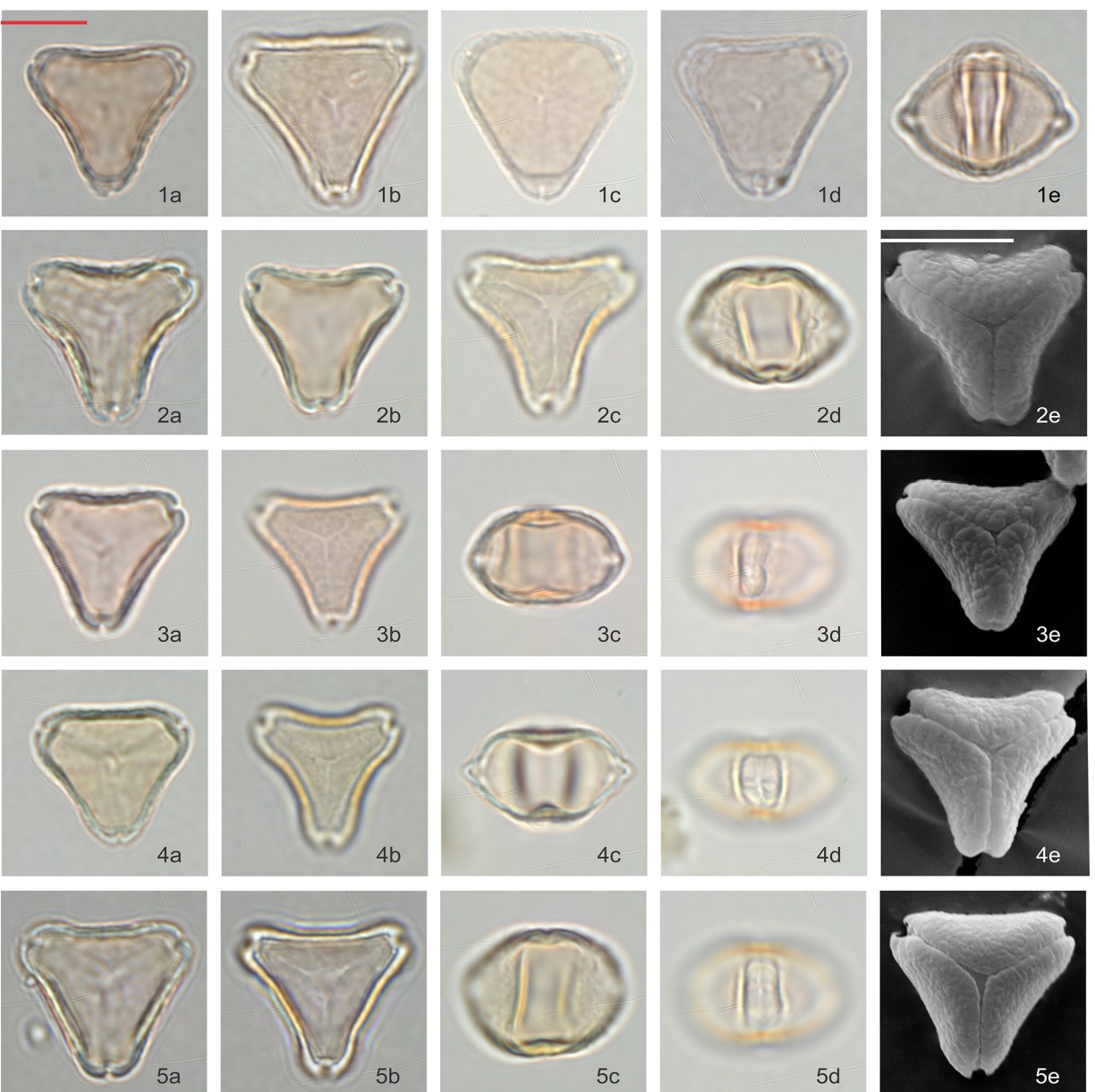

**Fig 5. Polar and equatorial views of *Leptospermum scoparium*, Part 3.** Scale bar = 10 μm. (1a-1e) Otaipango, (2a-2e) Northern South Island, (3a-3e) Wellington, (4a-4e) Papa, (5a-5e) Southern South Island.

wider range of pollen size is observed within "Auckland" type due to the larger number of specimens examined.

### *Kunzea* visual observations

Pollen grains are isopolar, oblate, triangular in polar view, and flattened oval in equatorial view. In polar view, the side of the amb appears to vary from slightly concave or straight, to slightly convex. Pollen is tricolporate and angulaperturate, very rarely di- or tetracolporate. Ectoapertures are usually narrow, and the endoapertures are narrow but lalongate.

Pollen grains are basically syncolpate, but more often appear to be slightly parasyncolpate, with small apocolpia at the poles. The exine appears uniformly psilate under LM and very

**Table 1. Pollen morphological characters of *Leptospermum scoparium* s.l. morphotypes (visual observation).**

| Morphotype | Size of Pollen Grains | | | | Characters of Pollen Grains | | | | | |
|---|---|---|---|---|---|---|---|---|---|---|
| | E (µm) | | P (µm) | | Sides of Amb | | Pattern | Vestibulum | Enlarged Apices | Thickened Arci |
| | Average | Range | Average | Range | Concave | Straight | | | | |
| Otaipango | 19.2 | 17.0–20.8 | 14.4 | 12.1–16 | | + | ++ | ++ | - | - |
| Papa | 19 | 15.9–20.9 | 11.6 | 10.6–13 | + | | ++,+ | ++ | + | + |
| Wellington | 17.9 | 15.7–20.4 | 11.5 | 10.3–12.6 | + | | ++ | +,- | - | - |
| Northern South Island | 19.1 | 18.0–20.9 | 14.2 | 13.1–15.3 | + | | ++ | - | + | + |
| Southern South Island | 19.1 | 18.4–21.6 | 14.1 | 12.9–16.4 | + | | + | - | + | - |
| Three Kings | 20 | 18.2–21.3 | 10.3 | 9.3–12.1 | | + | ++,+ | + | + | - |
| East Cape | 18.1 | 16–20.7 | 11.6 | 10–13.1 | ++, + | | + | - | + | + |
| Flat Silver | 19 | 16.9–20.3 | 13.4 | 12.7–14.1 | ++, + | | ++,+ | +,- | +,- | + |
| Surville Cliffs | 20.6 | 18.2–22.1 | 13.6 | 13.1–15.1 | ++ | | + | + | + | - |
| Central Volcanic Plateau | 18.5 | 16.6–20.1 | 13 | 11.1–14.3 | + | | + | - | + | + |
| *L. scoparium* var. *incanum* | 19.4 | 17.3–21.9 | 16 | 13.5–17.5 | + | | ++,+ | + | + | - |
| South Island Mountain | 19.8 | 17.9–22.2 | 14 | 12.6–15.8 | + | + | + | +,- | + | + |
| Waikato Peat Bog | 18.6 | 16.5–20.1 | 13.4 | 12.7–14.3 | + | + | ++,- | - | +,- | - |
| Auckland | 18.8 | 16–22.4 | 11.7 | 10.7–13.1 | + | + | + | - | + | - |
| Coromandel Swamp | 18.3 | 17.1–20.3 | 14.3 | 12.8–16.1 | + | + | + | - | + | - |

++ denotes feature obviously present

+ denotes feature present

- denotes feature not obviously seen (multiple scores indicate a range of feature states)

obscurely patterned under SEM (Figs 6 and 7). The size of P axis is 11.28 ± 2.33 µm (n = 103), and E axis 16.30 ± 0.95 µm (n = 318).

Of the ten species, *K. toelkenii* is the only species that could be separated from the others–by size–with mean E-axis of 17.8 µm compared with <17 µm for other *Kunzea* spp., along with slightly enlarged apices (Table 2, Fig 6). Of the remaining species, we could see no apparent difference in the size or surface texture. We do note some range of amb appearance in polar view, described in Table 2, but we observe considerable variability in all populations as well, such that we regard these observations a tendency, rather than a diagnostic feature. Given we do not observe consistent repeatable differences between the other species of *Kunzea*, we have not tabulated observations from the small number of hybrid combinations sampled.

Our size measurements of *Kunzea* pollen tend to be larger than those previously reported by de Lange [11]. This is likely due to the acetolysis treatment applied in this study, which tends to enlarge pollen grains [42, 43].

## Classifynder measurements of *Leptospermum scoparium* s.l and *Kunzea* spp.

Inspection of distribution histograms (raw scan data is included in S5 Table) revealed maximum Feret diameter and area are the two parameters (of the 50 parameters described above) that most clearly differentiate *Leptospermum scoparium* s.l. and *Kunzea* pollen. This is consistent with the observations of Holt and Bebbington [29], who conducted a similar study on four samples of *L. scoparium* s.l. and *Kunzea*. An overlap between the two genera was observed for both parameters (Fig 8).

Using the Classifynder, the 2.s.d. equatorial diameter of *L. scoparium* s.l. pollen is 17.93–22.18 µm (*n* = 11932), while the 2.s.d. equatorial diameter of *Kunzea* pollen is 14.32–19.00 µm

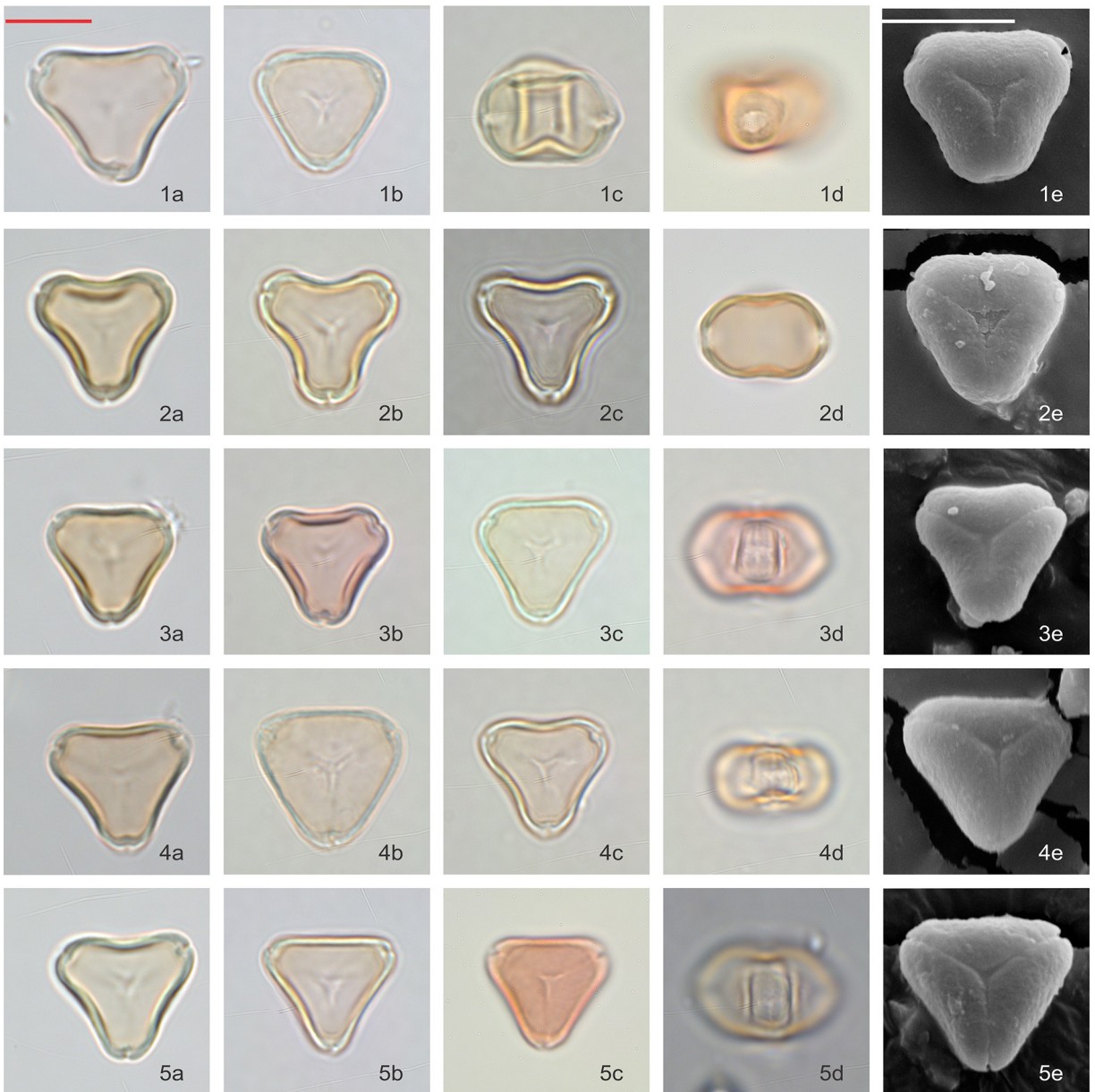

**Fig 6. Polar and equatorial views of *Kunzea*, Part 1.** Scale bar = 10 μm. (1a-1e) *K. toelkenii*, (2a-2e) *K. sinclairii*, (3a-3e) *K. serotina*, (4a-4e) *K. amathicola*, (5a-5e) *K. ericoides*.

(*n* = 3678), calculated from the maximum Feret diameter measurements (MFD) of polar view images. While the relative difference between *L. scoparium* s.l. and *Kunzea* was similar for both the visual measurements and the Classifynder, maximum Feret diameter from the Classifynder measurements is consistently 0.5–1 μm larger than light-microscope measurements (S1 and S2 Figs). One reason for this discrepancy is that for highly convex grains, the MFD measurement may not necessary be the equatorial diameter as defined in Fig 2, but a capture of a different axis, which is also observed by Holt and Bebbington [29]. This is supported by the observation that the smallest offsets between MFD and visual equatorial measurements are found in the *Leptospermum scoparium* morphotypes Otaipango, Flat Silver and Surville Cliffs,

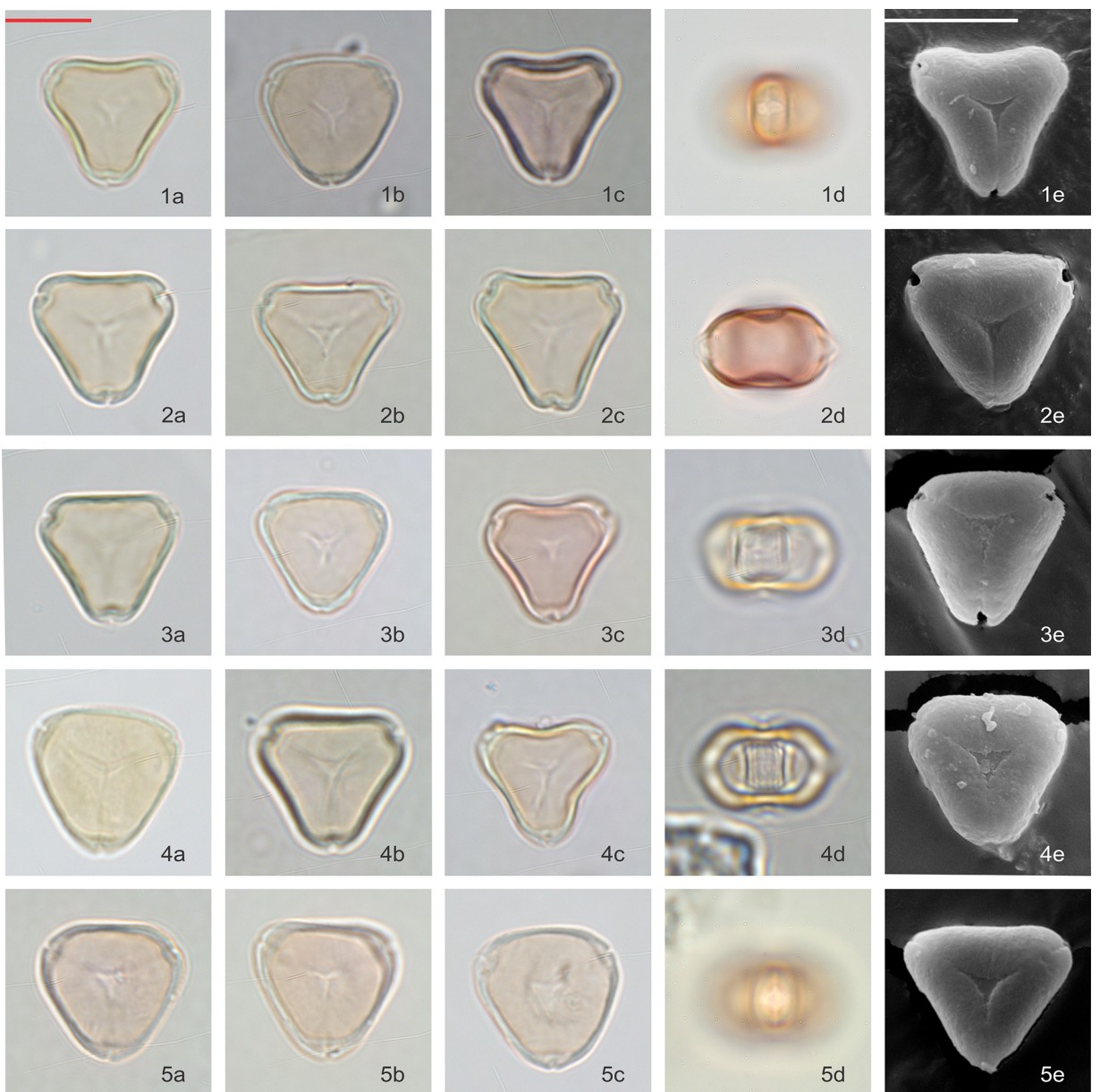

**Fig 7. Polar and equatorial views of *Kunzea*, Part 2.** Scale bar = 10 μm. (1a-1e) *K. linearis*, (2a-2e) *K. tenuicaulis*, (3a-3e) *K. salterae*, (4a-4e) *K. robusta*, (5a-5e) *K. triregensis*.

which have ambs either straight or slightly concave-sided, while relatively larger deviations are observed within convex-sided *Kunzea* species. We chose to rely on visual measurements of equatorial diameter for the taxonomic descriptions.

## Ordination of Classifynder measurements

Although Linear Discriminant Analysis (LDA) seeks to maximize class separation (i.e., between-class variance), LDA of our pollen data produced a very similar result to the 'unsupervised' ordination of Principal Components Analysis. Both these analyses show distinct groups of *L. scoparium* s.l. and *Kunzea* pollen in ordination space, but with overlap: we report on and illustrate the PCA results here.

**Table 2. Pollen morphological characters of *Kunzea* species (visual observation).**

| *Kunzea* species | Size of Pollen Grains | | | | Characters of Pollen Grains | | | | |
|---|---|---|---|---|---|---|---|---|---|
| | E (μm) | | P (μm) | | Sides of Amb | Pattern | Vestibulum | Enlarged Apices | Thickened Arci |
| | Average | Range | Average | Range | | | | | |
| *K. toelkenii* | 17.8 | 15.2–20 | 10 | 9.4–10.6 | slightly concave | - | - | + | - |
| *K. linearis* | 16.9 | 16.3–17.5 | 10.5 | 10–10.6 | slightly concave | - | - | - | - |
| *K. amathicola* | 16.8 | 15.6–18.1 | 9.6 | 8.8–11.3 | slightly concave | - | - | - | - |
| *K. triregensis* | 16.7 | 15.2–17.6 | 9.1 | 8.8–10 | slightly convex | - | - | - | - |
| *K. tenuicaulis* | 16.1 | 15.0–16.9 | 9.6 | 8.8–10.6 | straight | - | - | - | - |
| *K. robusta* | 16 | 15.0–17.5 | 9.9 | 9.4–10.6 | slightly concave | - | - | - | - |
| *K. ericoides* | 16.6 | 14.4–18.1 | 10.2 | 9.4–11.3 | slightly concave | - | - | - | - |
| *K. sinclairii* | 16.2 | 15.2–17.6 | 9.7 | 8.8–10 | more concave | - | - | - | - |
| *K. salterae* | 15.4 | 14.4–16.8 | 9.8 | 9.4–10.6 | straight | - | - | - | - |
| *K. serotina* | 15 | 13.8–16.3 | 9.4 | 8.8–10.6 | slightly concave | - | - | - | - |

\+ denotes feature present

\- denotes feature not obviously seen

For a PCA constrained by the 7 'core' parameters, 85% of the variance is described by the first two axes (Fig 9), while only 66% variance is described by the first two axes for a PCA constrained by all 50 parameters. Although there is overlap between *L. scoparium* s.l. and *Kunzea* pollen in the ordination, we see a clear differentiation between the two populations. In the ordination of the 7 'core' parameters, the most important for discriminating between *L. scoparium* s.l. and *Kunzea* are maximum Feret diameter, compactness, convex hull, and Heywood Circularity.

We explored the overlap of the morphotypes of *L. scoparium*, and the species of *Kunzea*, in subsequent PCA analyses. In both cases, we do not see strong differentiation between *L. scoparium* s.l. morphotypes (Fig 10), or *Kunzea* species (Fig 11).

## Differentiation of *Leptospermum scoparium* s.l. and *Kunzea* spp. by Support Vector Machine

The Support Vector Machine trial shows acceptable skill at discrimination between *L. scoparium* s.l. and *Kunzea* pollen in our dataset (Table 3). The best results are obtained where all 50 parameters are used to discriminate between specimens in polar view, with a mean prediction accuracy of 97.2%. Where only the seven core parameters are used, and the model is required to discriminate between specimens in both polar view and equatorial view, prediction accuracy drops to 94.7%.

## Discussion

### Generic discrimination

The results of this study confirm observations from previous work based on smaller sample sizes. The large sample size and comprehensive sampling approach in the present study allows differences between the pollen of the New Zealand representatives of *Leptospermum* and *Kunzea* to be identified with more confidence. Firstly, there is a significant difference in average size between pollen of the two genera. Our results suggest that it should be possible to estimate the relative proportions of the two genera in a mixed population from a statistical study of equatorial pollen dimensions. *Leptospermum scoparium* s.l. pollen (18.88–21.50 μm,

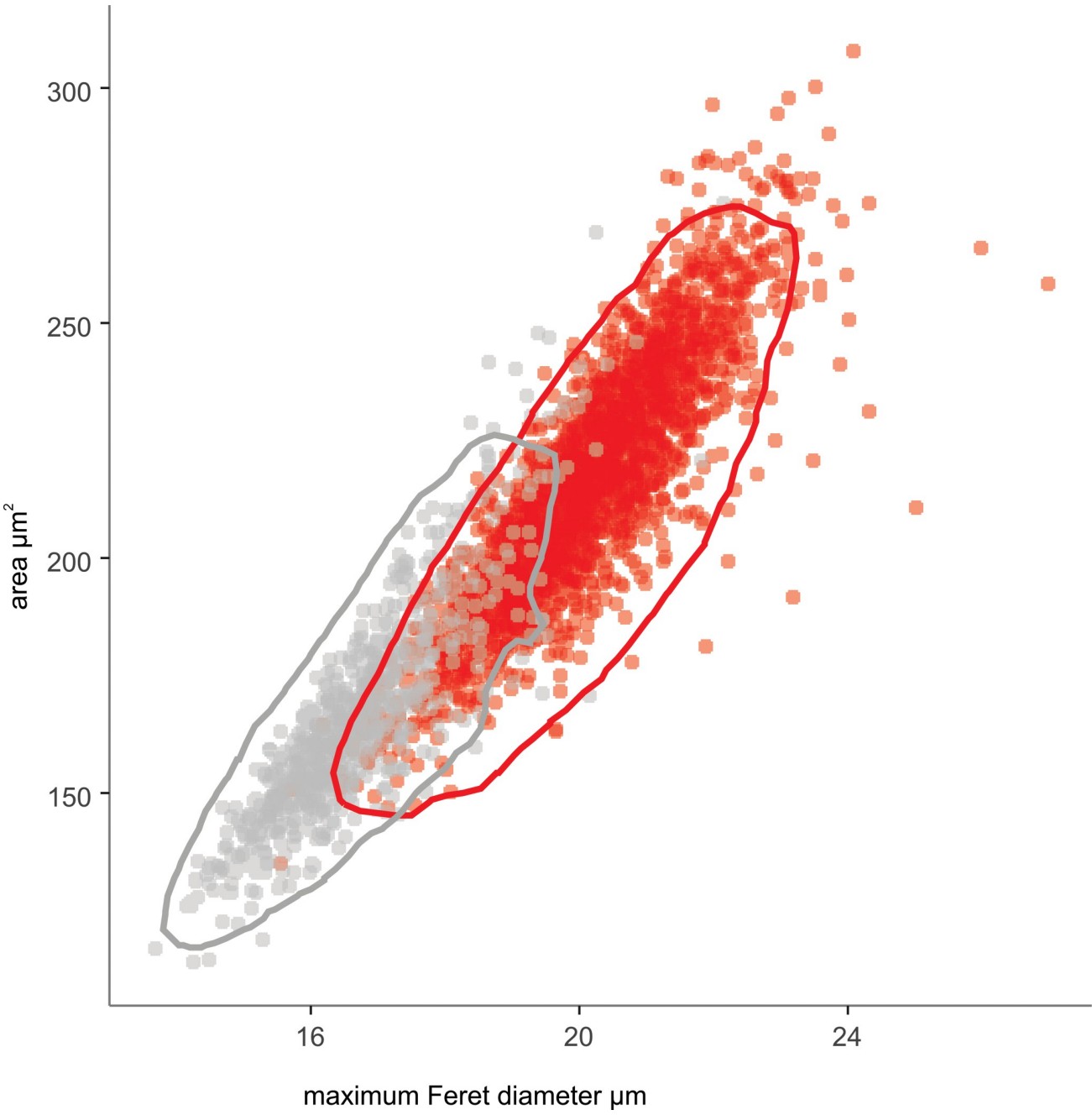

**Fig 8. Classifynder measurements of area vs maximum Feret diameter of pollen in polar view.** *Leptospermum scoparium* (red, n = 11932 specimens) and *Kunzea* (grey, n = 3678). Polygons show the 95% range of the highest posterior density region for each species, calculated using R package coda [44].

n = 11932) could be discriminated from *Kunzea* spp. pollen (15.49–17.83 μm, n = 3768) using the one standard deviation ranges of equatorial diameter. Only one species of *Kunzea*, *K.toelkenii*, has a pollen size comparable to that of the smallest pollen of *Leptospermum scoparium* morphotypes, e.g "Wellington" and part from "Auckland". However, specimens of this large *Kunzea* pollen should be easily separated from small *Leptospermum scoparium* s.l. specimens on their less concave amb, less patterned surface, and presence of apocolpia.

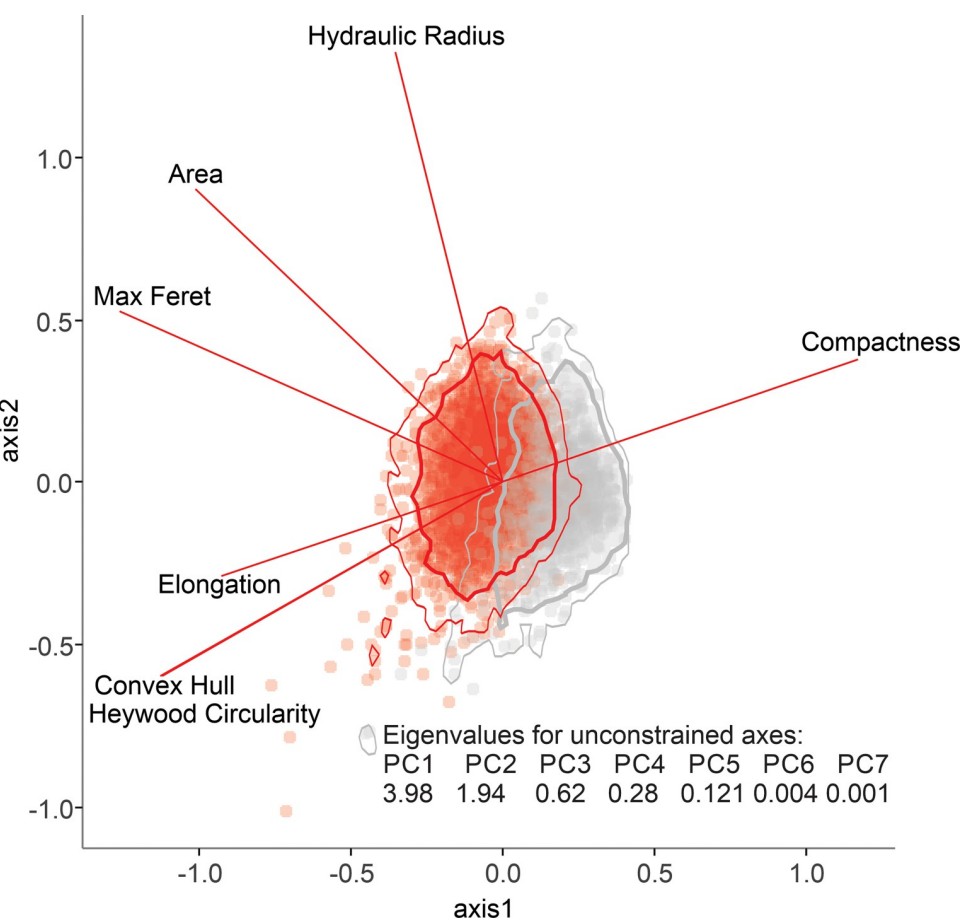

**Fig 9. Principal component analysis of the seven core parameters for *Leptospermum scoparium* and *Kunzea* pollen in polar view.** Polygons show the highest posterior density region for each species, thick lines = 95% range, thin lines = 99% range, calculated using R package coda [44].

In addition to the size, our observations indicate shape is another characteristic that allows discrimination of the two pollen types. Nearly all *Leptospermum scoparium* s.l. pollen is clearly concave-sided in polar view and more angular, while *Kunzea* pollen is generally nearly straight-sided. Pollen of *K. sinclairii* is slightly concave-sided, but with angles much less extended compared to *Leptospermum scoparium* s.l.. McIntyre [23] concluded that 55% of *Kunzea* pollen (as *L. ericoides*) have convex sides, compared with 4% of *Leptospermum scoparium* s.l. grains. It is hard to assess the range of the coverage of McIntyre's specimens, taking the new taxonomic revision of *Kunzea* into account [11]. Another observation by McIntyre [23], that the colpi are more frequently para-syncolpate in *Kunzea* than in *Leptospermum scoparium* s.l., is confirmed by the present study.

The surface sculpture of Australian *Leptospermum* and *Kunzea* pollen was discussed by Thornhill et al. [45]. In contrast to Thornhill et al. [45], who noted both scabrate and psilate exine patterning in Australian *Leptospermum* and *Kunzea* pollen, our study of New Zealand grains suggests exine pattern can be used to distinguish between the two. For *Leptospermum scoparium* s.l., the exine tends to be more coarsely sculptured (i.e., scabrate), while for *Kunzea* pollen, the exine is less patterned and appears to be more psilate. While exine of *L. scoparium* s.l. exhibits different levels of coarseness, it is a feature that is consistently present through all specimens examined, and is clearly resolved under LM and prominent under SEM. In contrast, the exine of *Kunzea* pollen is consistently nearly psilate.

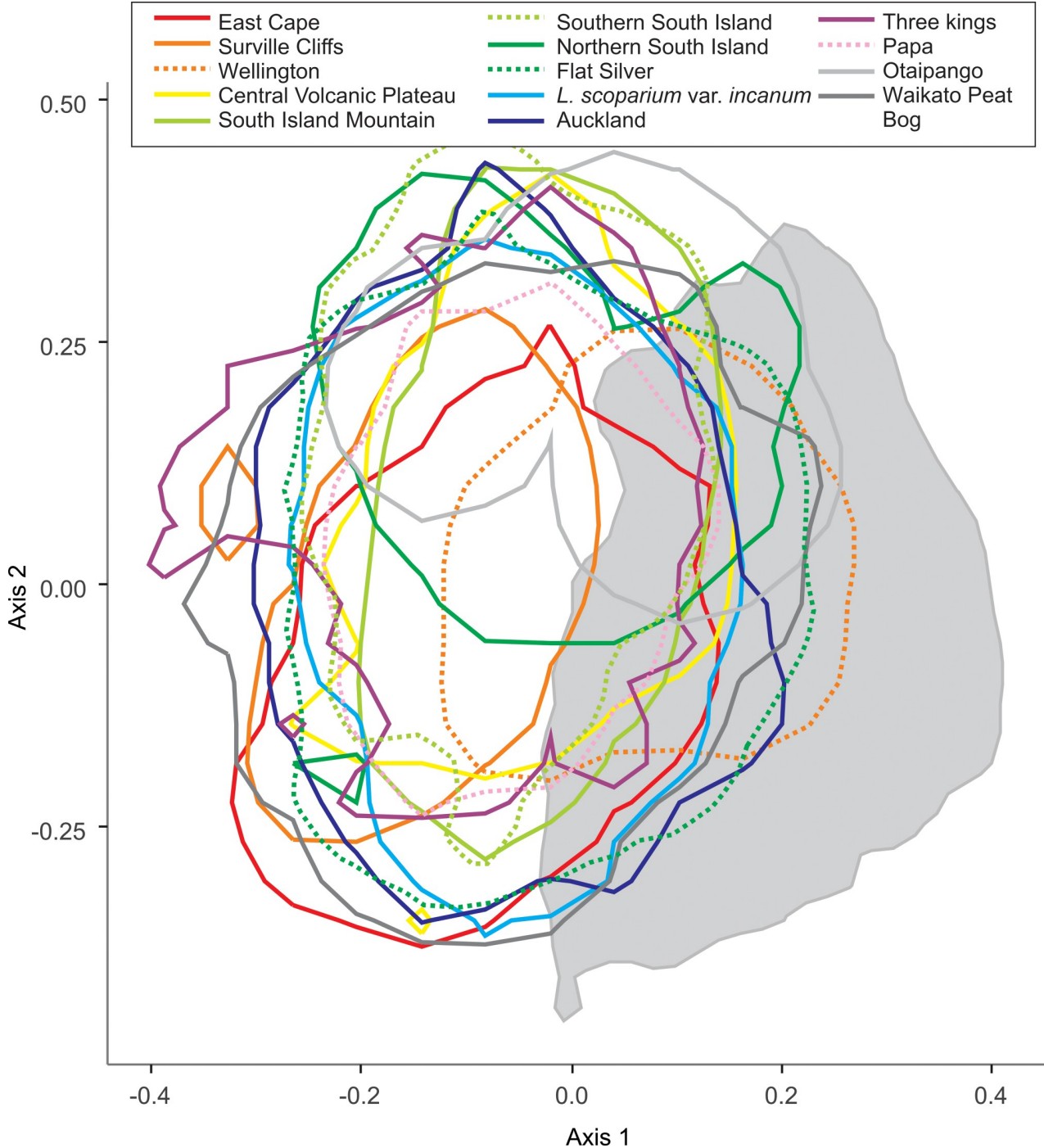

**Fig 10. Principal component analysis of the seven core parameters for *Leptospermum scoparium* morphotypes pollen in polar view.** Polygons show the 95% range highest posterior density region for each sub species. Underlying grey polygon is 95% range of *Kunzea* from Fig 9.

## Variability within New Zealand *Leptospermum* and *Kunzea*

Although we see a clear distinction between pollen of *Leptospermum scoparium* s.l. and *Kunzea* spp., differences in pollen between the morphotypes of *Leptospermum scoparium* s.l. and between species of *Kunzea* are more subdued. For *Kunzea*, the average pollen size, combined

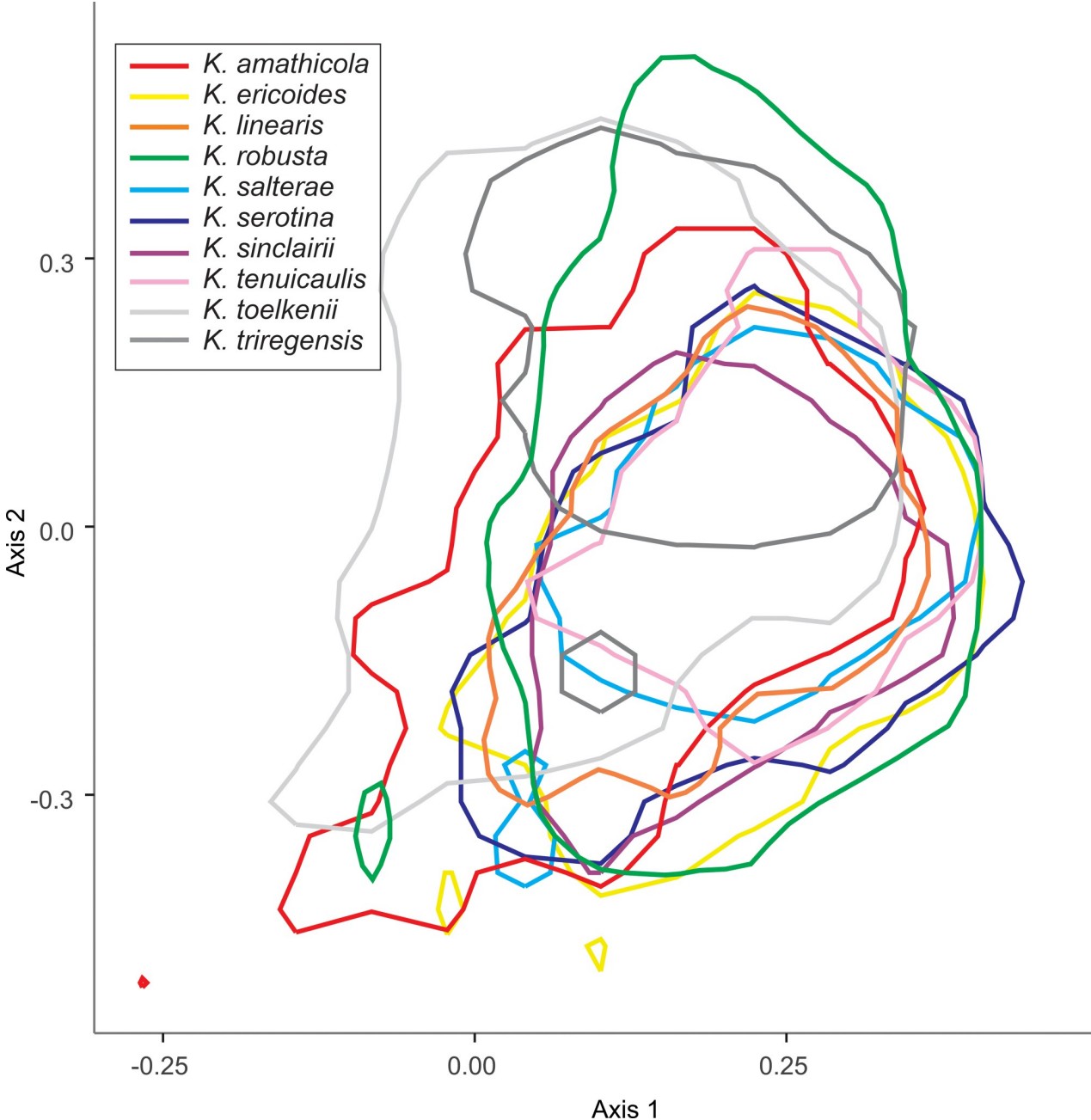

**Fig 11. Principal component analysis of the seven core parameters for *Kunzea* species pollen in polar view.** Polygons show the 95% range highest posterior density region for each sub-species.

with concaveness of amb and feature of apices, make it possible to separate *K. toelkenii* from the others under LM (Figs 6 and 12). Despite these noticeable characteristics, taking intraspecific variation into account, pollen of species within *Kunzea* spp. are practically indistinguishable from one another. For *Leptospermum scoparium* s.l., our initial results suggest that some morphological characteristics observed under LM, including size, texture, concaveness of amb, and presence of vestibulum, will be useful to characterise groups of morphotypes. However, Classifynder measurements made in this study do not support differentiation within species of

**Table 3. Cross-validation results of Support Vector Machine to predict pollen type, based on 99 iterations of random 80:20 splits of data into training:test sets.**

| Model parameters | precision rate ± s.d. |
| --- | --- |
| 50 parameters, polar view and equatorial view | 96.1% ± 0.3 |
| 50 parameters, polar view only | 97.2% ± 0.3 |
| 7 parameters, polar view and equatorial view | 94.7% ± 0.3 |
| 7 parameters, polar view only | 95.9% ± 0.3 |

*Kunzea* or morphotypes of *Leptospermum scoparium*. This could be partly due to limitations of the apparatus (as discussed below), partly to intraspecific pollen variation within each entity.

Intraspecific pollen variation is found in many of the Myrtaceae, but separation of 'true' variability from confounding issues of experiment design, such as variability in sample processing techniques or environmental morphotypes, remains a major challenge and a matter of debate [46]. Some types of morphological variations, such as the presence of four apertures (especially frequently presented in hybrids), are likely due to natural factors (e.g., genetic variation), as noted in some Australian Myrtaceae species [46]. Although a few aberrations, e.g. where a single flower produces two morphologically different pollen grains, are consistently found in some species of *Eucalyptus* and occasionally in other Myrtaceae species [46], this is rare or not apparent in morphotypes of *Leptospermum scoparium* s.l. we have examined. However, other types of morphological variation, for example variability in size, or shape, may reflect a combination of true variability and experimental treatment.

Pollen size can be affected by chemical treatment and mounting medium, maturity of grains, and pollen preservation [43, 47–51]. Although pollen grains shrink under dry and expand under wet conditions, we do not expect freshness of herbarium specimens to be a significant contributor to the intra-specific size variation observed, because shrinkage caused by dehydration is reversible [43]. In addition, because we have applied consistent preparation procedures, chemical treatment and mounting medium are not likely to be the cause of within-group variation in grain size. We do consider it likely that intraspecific size variability could arise from different levels of pollen maturity, along with other factors such as variable sporopollenin content, as discussed by Adeleye et al. [46]. In their study, a small proportion of larger, thinner-walled and convex-sided pollen grains were observed along with "normal" concave-sided grains from the same individual specimen of *Leptospermum*. This diversity is also observed in some of the samples we analysed and may be partly due to variation in exine development at different levels of maturity of pollen sampled from multiple anthers of the same specimen.

## Utility of the Classifynder

The relative ease of collection of a large dataset, and the skill of the Support Vector Machine at discrimination between *Leptospermum scoparium* s.l. and *Kunzea* spp. pollen (~95%), shows promise for routine application of the Classifynder or similar device. In some respects we find this level of skill surprising, because some of the characteristics important for a human observer using a 100x oil immersion objective to discriminate between the two types are not available to the lower-resolution Classifynder. In particular, we expect the surface sculptural elements of pollen grains from *Leptospermum scoparium* s.l. and *Kunzea* spp., normally less than 1 μm, are of sizes at or below the limit of resolution of the digital imaging used in the Classifynder, which is approximately 1 μm [52]. More generally, we speculate that there may be potential to further improve the skill of a Classifynder-like machine for narrow two-class

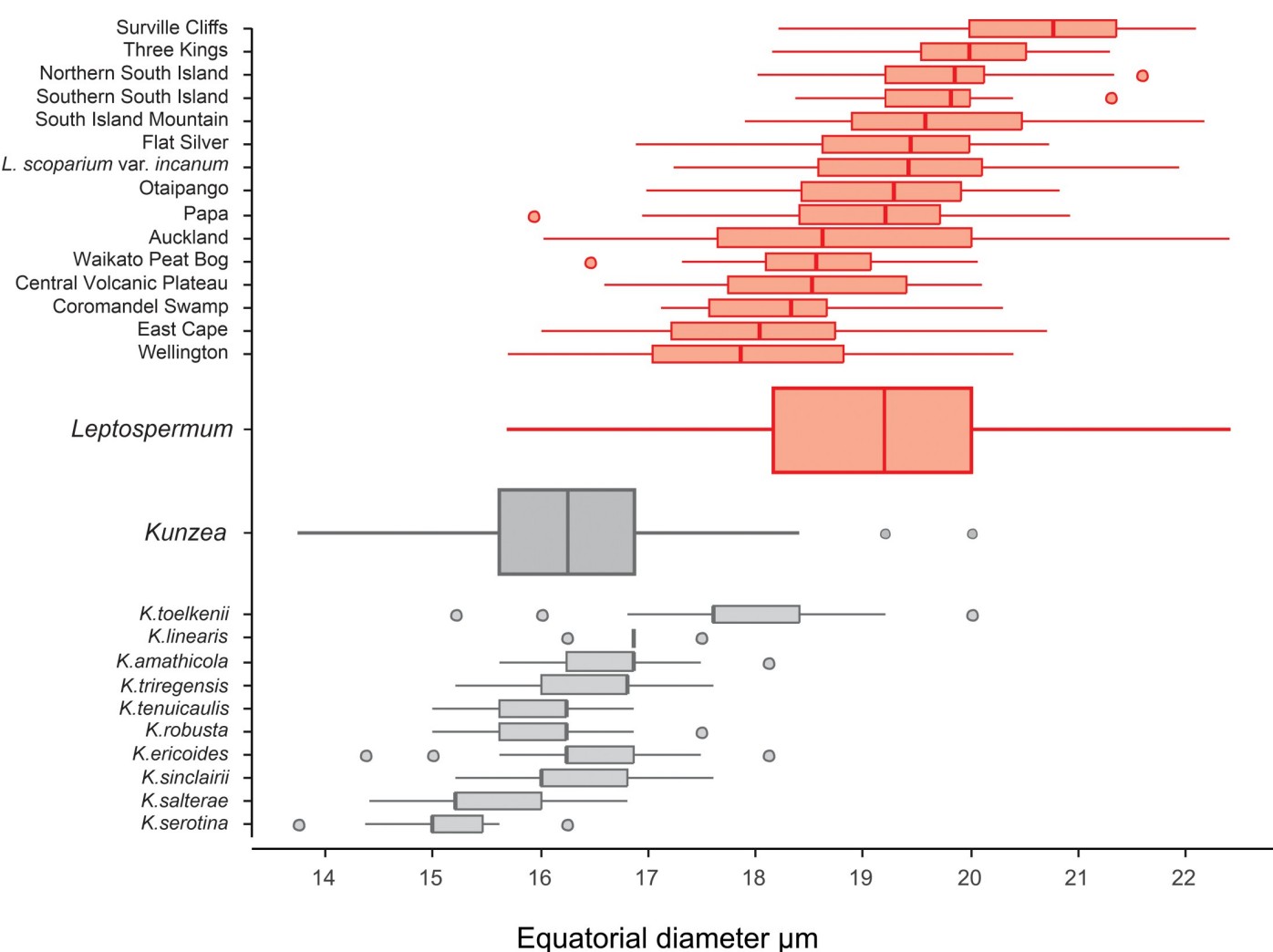

**Fig 12. Size range of *Leptospermum scoparium* and *Kunzea* in equatorial diameter under light microscope.** *Leptospermum scoparium*–upper, red; *Kunzea*–lower, grey.

pollen discrimination problems such as this one. This is because the 50 features captured by the Classifynder were selected during machine design to allow discrimination of a wide range of pollen morphology, with a particular focus on 5 disparate end-member forms [37]. It is possible that a machine optimised for discrimination between a smaller range of pollen forms would be built to capture a different array of features.

Over the past decade, the rapid increase in capability in computational intelligence has resulted in considerable achievements in automatic pollen identification. Deep Learning Convolutional Neural Networks (DLCNN) is one of the most promising applications in the field of image recognition for pollen identification. Compared to a traditional neural network, DLCNN contains many more layers of neurons and is thus appropriate to very large data sets. Based on its capability to process and filter important signals from each layer of complex artificial neurons, DLCNN could extract the features that the network determines to be the most discriminatory aspects of the classes provided in the training sets. This process of feature identification is automatic and free from manual intervention, which obviates the procedure of manually selecting polar and equatorial orientation of pollen grains as practised in this study,

while increasing reliability. As a result, over 97% accuracy on a set of 23 different pollen types from Brazil has been achieved by applying Deep Learning Convolutional Neural Networks (DLCNN) to a pollen image dataset composed of 805 microscope images [53]. A further exploration of DLCNN using a larger dataset of over 19,000 pollen images generated using the Classifynder produced even more promising results [54], with a classification success rate up to 98% across 46 different pollen types including New Zealand *Leptospermum scoparium* and *Kunzea ericoides*. Further improvements are likely, through increased sample sizes, refined measurement criteria, and improved discrimination networks. The deployment of the Classifynder for image capture and application of DLCNN for pollen classification could be a potential solution.

## Conclusion and next steps

We have demonstrated that consistent differentiation between *Leptospermum scoparium* s.l. and *Kunzea* spp. pollen collected from New Zealand herbarium specimens is possible, using both visual and Classifynder light microscope measurements. The key characteristics that differentiate *L. scoparium* and *Kunzea* pollen are size, shape of amb, and surface texture. A Support Vector Machine trial based on the Classifynder measurements showed acceptable skill at discrimination between *L. scoparium* and *Kunzea* pollen. The best results were obtained when all 50 Classifynder parameters were used to discriminate between specimens in polar view, with a mean prediction accuracy of 97.2%.

While there is some indication that differentiation between pollen from some morphotypes of *L. scoparium* may be possible, further work on a larger sample set (preferably in conjunction with taxonomic studies) would be required to confirm this. An exploration of DLCNN on the larger dataset might have potential to improve the understanding of this issue.

These results could provide a first step to applying melissopalynology techniques to determine the relative contributions of *L. scoparium* s.l. and *Kunzea* nectar in manuka honey. Further analysis of honey samples would be required to determine if there are significant correlations between bioactivities and certain types of *L. scoparium* pollen. Application to paleoecological studies and the geological history of Myrtaceae in New Zealand are other fields yet to be explored.

## Supporting information

**S1 File. Comparison of pollen dimensions of male and bisexual flowers of *Leptospermum scoparium* s.l. and *Kunzea robusta*.**
(DOCX)

**S1 Fig.** Comparison for *Leptospermum scoparium* (red) and *Kunzea* (grey) of equatorial diameter measured by palynologist using a light microscope, and maximum Feret diameter measured by Classifynder.
(TIF)

**S2 Fig.** Comparison for *Leptospermum scoparium* (upper) and *Kunzea* (lower) of equatorial diameter measured by palynologist using a light microscope (denoted by suffix "_h"), and maximum Feret diameter measured by Classifynder (denoted by suffix "_d").
(TIF)

**S1 Table. Herbarium specimens studied–*Leptospermum*.**
(XLSX)

**S2 Table. Herbarium specimens studied–*Kunzea*.**
(XLSX)

**S3 Table. Pollen measurements of *Leptospermum scoparium* s.l.**
(XLSX)

**S4 Table. Pollen measurements of *Kunzea*.**
(XLSX)

**S5 Table. Pollen features extracted by Classifynder including 7 core parameters and 43 additional parameters.**
(XLSX)

## Acknowledgments

We thank Ewen Cameron of Auckland Museum Herbarium (AK) for permission to destructively sample *Kunzea* and *Leptospermum* specimens under his care, and Dallas Mildenhall and Giuseppe Cortese for reviewing the 'paper before submission. Andrew Boyes assisted with drafting Fig 1. We also thank two external reviewers for their constructive comments.

## Author Contributions

**Conceptualization:** X. Li, J. G. Prebble, J. I. Raine.

**Data curation:** X. Li, J. G. Prebble, P. J. de Lange.

**Formal analysis:** J. G. Prebble.

**Investigation:** X. Li, J. G. Prebble, J. I. Raine.

**Methodology:** X. Li, J. G. Prebble, J. I. Raine.

**Resources:** P. J. de Lange, L. Newstrom-Lloyd.

**Supervision:** J. I. Raine.

**Writing – original draft:** X. Li, J. G. Prebble, P. J. de Lange, J. I. Raine.

**Writing – review & editing:** X. Li, J. G. Prebble, P. J. de Lange, J. I. Raine, L. Newstrom-Lloyd.

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
