## [Decision Letter · Decision Letter 0]

8 Mar 2022

PONE-D-21-39233Discrimination of pollen of New Zealand mānuka (Leptospermum scoparium agg.) and kānuka (Kunzea spp.) (Myrtaceae)PLOS ONE

Dear Dr. Li,

Thank you for submitting your manuscript to PLOS ONE. After careful consideration, we feel that it has merit but does not fully meet PLOS ONE’s publication criteria as it currently stands. Therefore, we invite you to submit a revised version of the manuscript that addresses the points raised during the review process.

The manuscript is rather longish and should therefore be streamlined. Reviewer #2 suggests focusing essentially on the goal, which was evidently to determine whether there are differences between Leptospermum and Kunzea pollen. Subtleties, such as the identification of different morphospecies, could be shifted to the supplement. Furthermore, it should be discussed what advantages the use of pollen analysis could have over DNA sequencing, for example when examining questionable honey samples. You can find further suggestions in the two reports by the Expert Reviewers.

We look forward to receiving your revised manuscript.

Kind regards,

Wolfgang Blenau

Academic Editor

PLOS ONE

Journal Requirements:

“Sampling and morphological study was funded by Kiwinet and GNS Science SSIF, while an initial study of pollen of male and hermaphrodite flowers was funded by New Zealand Ministry for Primary Industry.”

“XL received funding [no serial number] from KiwiNet (https://www.mbie.govt.nz/science-and-technology/science-and-innovation/funding-information-and-opportunities/investment-funds/preseed-accelerator-fund/kiwi-innovation-network-limited/) and GNS Science SSIF. The funder had no role in study design, data collection and analysis, decision to publish, or preparation of the manuscript.”

3. We note that you have referenced (P.J. de Lange unpubl.) which has currently not yet been accepted for publication. Please remove this from your References and amend this to state in the body of your manuscript: (ie “de Lange et al. [Unpublished]”) as detailed online in our guide for authors

Reviewers' comments:

Reviewer's Responses to Questions

**Comments to the Author**

1. Is the manuscript technically sound, and do the data support the conclusions?

Reviewer #1: Yes

Reviewer #2: Yes

2. Has the statistical analysis been performed appropriately and rigorously? 

Reviewer #1: Yes

Reviewer #2: Yes

3. Have the authors made all data underlying the findings in their manuscript fully available?

Reviewer #1: Yes

Reviewer #2: Yes

4. Is the manuscript presented in an intelligible fashion and written in standard English?

Reviewer #1: Yes

Reviewer #2: Yes

5. Review Comments to the Author

Reviewer #1: Thank you for the opportunity to review this manuscript. The authors’ work into morphological variation between pollen of mānuka and kānuka serves to reinforce previous work which has demonstrated that these two taxa can be separated based on pollen morphology. This has important implications for both quality control in the honey market, as well as in pollen-based vegetation reconstructions in New Zealand.

The paper is well-written and the experimental design sound. I have no hesitation in recommending that it be accepted for publication, subject to a few minor changes/suggestions which I detail below.

Line 16 (Abstract): “..as separation of pollen”. This would be more clearly put as “..as differentiation of these taxa..” or similar. “As separation of pollen” is too general/vague.

Line 40: a close-bracket appears to be missing here.

Line 41: May be useful to specify “New Zealand Government” before Ministry. Although the use of ‘Ministry’ may well imply that to a sufficient degree.

Lines 48 – 49: would be useful to have a reference for these variations in L. scoparium

Lines 50 – 52: this sentence is a bit awkwardly worded and clunky. Probably don’t need both “Nevertheless” and “despite this evidence”. Plus the inclusion of the physical description of one of the varieties in the sentence makes it hard to follow. Suggest reword to “ Nevertheless, until recently only one species, comprising two varieties, has been accepted for New Zealand. These two varieties are …..” etc.

Line 54: add comma after ‘writing’

Line 62: Suggest replace “NZ Quaternary environmental studies” with “NZ palaeovegetation studies”.

Line 69: Should be “Ministry for Primary Industries”.

Lines 86 – 87: More of a comment/suggestion here. Given there is already a standard for manuka honey (cf. lines 40 – 43), is there a need for a pollen-based standard? What value could this bring above the existing test? I think it would add to the impact of the paper if the authors could add a sentence or two which proposes why a standard including pollen morphology/melissopalynology could be beneficial for the industry. I’m aware that there are limitations to the existing MPI standard.

Line 94: I don’t really follow what is meant by “each taxonomic entity geographically apart”. Is it possible to make the meaning clearer here?

Lines 176 – 180: I’m a little bit confused as to how this could happen if the images were manually selected, i.e. I would have assumed that images selected would have been those which were not only properly orientated, but also appeared to be of good quality, i.e., not fuzzy, no debris adhering to the pollen, no hollowing out by aggressive edge detection etc. I’m assuming that some fault in the image is what is meant by “imperfections”.

Perhaps a better/briefer way to capture this aspect of the methodology is to state that only properly oriented images and good quality images (i.e. images without artifacts, etc.) were selected for use.

Alternatively, if the preference is to maintain the text as is, then it would be helpful to know more about the nature of the imperfections in the image quality and why compactness factor was used to define what was retained/discarded.

Line 308/333: a reference for the Coda package should probably be provided, i.e. Plummer et al (2006)

Lines 444 – 457: Yes, I totally agree overall. Yes, the native feature set of the Classifynder is not optimised to distinguishing pairs or sets of very similar types. The DLCNN approach applied in the papers referenced in this section (48 & 49) is probably the most promising avenue, as these NNs essentially define their own features based what they determine to be the most discriminatory aspects of the classes, as provided in the training sets. This aspect might we worth adding to the discussion. Likewise, in [49], the images of manuka and kanuka were randomly oriented, rather than polar & equatorial, and yet the system was still able to differentiate them reasonably well. I know from experience that sifting through 1000s of images for correctly aligned ones is time consuming and would really limit the deployment of the Classifynder in the routine application of a manuka pollen standard. Therefore, the ability to use randomly oriented pollen images is probably necessary here, thus making DLCNNs the best option. Again, I suggest the authors consider rewording the discussion to make reference to these issues.

Reviewer #2: Dear Xun and co-authors,

You have done a thorough job in examining pollen of New Zealand Leptospermum and Kunzea. What I found though was your manuscript lost focus on what your aim was. My understanding of your manuscript is that you want to find pollen characters that can help discriminate between the two NZ Leptospermeae species so that you can identify if the honey is Manuka honey or not. I think you have done that in this paper and it has to do with pollen measurements. But that message gets lost with the deep discussion you make on the differences within each species/morphospecies. I think the differences that you identify within the species are nothing major too. I think that most of the inter-species information could be moved to supplemental data or removed completely. Focus on the differences between the genera.

By focusing on the genera you also need to consider a couple of things.

In your introduction you say "in December 2017 the New Zealand Ministry for Primary Industries (MPI) chose a purity test for manuka honey based on assay of characteristic chemical marker compounds and DNA of manuka pollen". If this is the government policy for determining manuka honey, then why are you trying to identify pollen characters that would help identify manuka honey? Could you not say that DNA analysis of honey would unequivocally tell you if Leptospermum scoparium is in the honey? Why do palynology analyses? Could it be quicker and cheaper than a DNA test. If yes then say so.

This paper suggests that each genus probably has more than one NZ species. I'm not question that here. However, if there is more than one species in Leptospermum then does their pollen all produce manuka honey? If not then which ones. That would make identifying manuka honey using pollen even trickier.

One last consideration is that this study only used native NZ plants from herbarium specimens. You can correct me if I am wrong, but aren't there now Australian species of Leptospermum naturalised in NZ? How would a real world example to identify manuka honey work if you also have Australian Leptospermum pollen muddying the honey. How would you unequivocally say that the naturalised Leptospermum pollen wasn't in the honey?

If you can address these issues in the manuscript then I think it will be more impactful and of more use in honey pollen identification.

Below are comments made while reading the manuscript.

Regards,

Andrew Thornhill

Page 2 Line 14 - I think species needs to be changed to taxa to make grammatical sense. As it reads now it is "Myrtaceous species

Leptospermum scoparium and Kunzea spp (i.e. species)"

Page 2 Line 19 - Is 'a' needed before Classifynder?

Page 2 Line 37 - 'scoparium' isn't italicised

Page 2 Line 38 - Technically this statement is true in that Kunzea is the closest relative to Leptospermum in NZ. I wouldn't say Kunzea is the sister genus of Leptospermum however because phylogenies show them to be separated by at least Asteromyrtus and Neofabricia. See the phylogenies of Maurin et al 2021 and Thornhill et al 2015.

Page 5 line 87 - Do all morphotypes of Leptospermum scoparium form Manuka honey? If they don't, what is the point of looking for different types within Leptospermum scoparium?

Page 12, line 242 - This paper's goal is to determine whether the plants that make manuka honey have different pollen to those that don't. The problem I see here is that if there are 10 species of Leptospermum in NZ - do all of these species form pollen that makes manuka honey. If they don't then which Leptospermum species/morphotypes do? Then the next question is do those Leptospermum species have unique pollen?

Page 12, Line 248 - I would be a bit cautious in using pollen size to help define different species. It looks like the pollen size differences between your different morphospecies is 2um if that. If I was to gather the same pollen and measure I have a feeling my measurements would have a greater error margin than 2um. The ability to recreate your pollen measurements means that everyone would need to use the same magnification, the same eye lens and the same microscope. Given that your size differences between plants is so small I think it is hard to justify the differences.

Page 11 Table 1. How can a feature be obviously seen & not seen at the same time? I'm confused by this table

Page 14, line 286 - This sentence highlights why you need to be cautious using pollen in taxonomy. If you don't use the exact same processing methods and microscopes then you are likely to get a different measurement. The difference in the same species between Pete's observations and the current one is around 3-4 um. This study shows that morphotypes have a difference of 1um or less. If a third study was done they would need to sample all morphotypes to come up with a range of smallest to largest. You wouldn't be able to sample just one morphotype and confidently conclude that you have X morphotype.

Page 15 line 310 - If the goal of the paper is to show the difference between Leptospermum and Kunzea pollen, then I think it appears quite simple. If a pollen is 14.32-17.93 in equatorial diametre then it is Kunzea. If it is 19-22.18 then it is Leptospermum. Anything in between can not be confidently assigned to a genus. The same would apply for measurements in polar view. It's the extreme of each measurement where there is no overlap between the two genera that are the unique pollen feature.

Page 17 line 362. Extra comma before full stop at end of sentence 1.

Page 18 line 381-382. If they can be separated by 1 St. Dev., why would you use another measure that is less accurate?

Page 19 line 396 - I think this is the incorrect citation. It should be this one -

Thornhill, A. H., Wilson, P. G., Drudge, J., Barrett, M. D., Hope, G. S., Craven, L. A., et al. (2012). Pollen morphology of the Myrtaceae. Part 3: tribes Chamelaucieae, Leptospermeae and Lindsayomyrteae. Aust. J. Bot. 60, 225–259. doi:10.1071/BT11176.

Page 19 line 412. Were they able to be separated or not? If there's enough variation within each that they can't be separated, does that not suggest they are a conglomerate?

Page 20 line 424 Is there a range of variation that encompasses this? What is the level of distinction that you feel demonstrates the shift from intraspecific to interspecific variation?

Page 20 line 436 Do your separated populations interbreed at all? Are there other Leptospermum spp. or closely related species?

Page 21 line 455. If Classyfinder has such a high accuracy, then it should have been able to detect any valid differences between the pollen morphologies in the dataset.

Page 21 line 459. Has this been demonstrated effectively? The paper seems to have lost focus on this purpose. The interest in distinguishing the Leptospermum scoparium populations is clear, but the abstract doesn't seem to accurately represent the paper's contents.

6. PLOS authors have the option to publish the peer review history of their article (what does this mean?). If published, this will include your full peer review and any attached files.

Reviewer #1: **Yes: **Katherine Holt

Reviewer #2: **Yes: **Andrew Thornhill

---

## [Author Response · Author response to Decision Letter 0]

22 Apr 2022

Editor’s comments

The manuscript is rather longish and should therefore be streamlined. Reviewer #2 suggests focusing essentially on the goal, which was evidently to determine whether there are differences between Leptospermum and Kunzea pollen. Subtleties, such as the identification of different morphospecies, could be shifted to the supplement. Furthermore, it should be discussed what advantages the use of pollen analysis could have over DNA sequencing, for example when examining questionable honey samples. You can find further suggestions in the two reports by the Expert Reviewers.

------Our goal was to explore the range of variation in pollen morphology displayed in New Zealand Leptospermum and Kunzea. To make this useful for melissopalynological characterisation of NZ honeys, this required examination of specimens of all formally and informally established species-level segregates across their geographic range. The results could also be relevant to paleoecological studies, and perhaps to taxonomy. 

The text relating to possible differences at intrageneric level, within the Results and Discussion sections of the manuscript, are therefore an essential part of the paper. We have carefully considered if these can be transferred to a supplement, and consider this would be both difficult and a detriment to readership value.

We have revised the Introduction to briefly explain the advantages of pollen analysis. The DNA analysis used in the NZ Government export requirement for honey designated as “manuka monofloral” or “manuka multifloral” is not quantitative (beyond establishing a non-trivial presence of Leptospermum pollen) – the classification is mainly based on the level of a single trace biochemical attributed to Leptospermum. The validity of this test in terms of relative manuka nectar contribution to the honey has been disputed. This is a large topic which we cannot adequately address in our paper, and which needs to be further researched. Our discrimination of NZ Leptospermum and Kunzea pollen provides one method for such research.

We have attended to the reviewer’s questions and comments as noted below. We found these very helpful. Resulting amendments to the text, and inclusion of additional references have tended to lengthen the paper, but we have sought to reduce the length and increase focus by eliminating some sub-sections where there was a degree of duplication or lesser relevance. The main cuts are in the original lines 242-251, 273-296 (Fig 8 moved to Discussion), and 353-373; also Table 2 has been reduced by omitting de Lange’s previous measurements.

The resulting text is in the end shorter than that originally submitted and we believe substantially improved. We hope you will agree.

Journal Requirements:

“Sampling and morphological study was funded by Kiwinet and GNS Science SSIF, while an initial study of pollen of male and hermaphrodite flowers was funded by New Zealand Ministry for Primary Industry.”

“XL received funding [no serial number] from KiwiNet (https://www.mbie.govt.nz/science-and-technology/science-and-innovation/funding-information-and-opportunities/investment-funds/preseed-accelerator-fund/kiwi-innovation-network-limited/) and GNS Science SSIF. The funder had no role in study design, data collection and analysis, decision to publish, or preparation of the manuscript.”

------ We have removed the funding acknowledgments from the manuscript and would like the Funding Statement to be amended to the following:

“An initial study of pollen of male and hermaphrodite flowers by JIR was funded by New Zealand Ministry for Primary Industry. XL received funding [no serial number] from KiwiNet (https://www.mbie.govt.nz/science-and-technology/science-and-innovation/funding-information-and-opportunities/investment-funds/preseed-accelerator-fund/kiwi-innovation-network-limited/) and GNS Science SSIF. The funders had no role in study design, data collection and analysis, decision to publish, or preparation of the manuscript.”

3. We note that you have referenced (P.J. de Lange unpubl.) which has currently not yet been accepted for publication. Please remove this from your References and amend this to state in the body of your manuscript: (ie “de Lange et al. [Unpublished]”) as detailed online in our guide for authors.

------This has been corrected.

 

Review Comments to the Author

Reviewer #1: Thank you for the opportunity to review this manuscript. The authors’ work into morphological variation between pollen of mānuka and kānuka serves to reinforce previous work which has demonstrated that these two taxa can be separated based on pollen morphology. This has important implications for both quality control in the honey market, as well as in pollen-based vegetation reconstructions in New Zealand.

The paper is well-written and the experimental design sound. I have no hesitation in recommending that it be accepted for publication, subject to a few minor changes/suggestions which I detail below.

Line 16 (Abstract): “..as separation of pollen”. This would be more clearly put as “..as differentiation of these taxa..” or similar. “As separation of pollen” is too general/vague.

------Accepted and change made

Line 40: a close-bracket appears to be missing here.

------Corrected

Line 41: May be useful to specify “New Zealand Government” before Ministry. Although the use of ‘Ministry’ may well imply that to a sufficient degree.

------Accepted and change made

Lines 48 – 49: would be useful to have a reference for these variations in L. scoparium.

------References inserted

Lines 50 – 52: this sentence is a bit awkwardly worded and clunky. Probably don’t need both “Nevertheless” and “despite this evidence”. Plus the inclusion of the physical description of one of the varieties in the sentence makes it hard to follow. Suggest reword to “ Nevertheless, until recently only one species, comprising two varieties, has been accepted for New Zealand. These two varieties are …..” etc.

------Reworded

Line 54: add comma after ‘writing’

------comma added

Line 62: Suggest replace “NZ Quaternary environmental studies” with “NZ palaeovegetation studies”.

------Accepted and change made

Line 69: Should be “Ministry for Primary Industries”.

------Corrected

Lines 86 – 87: More of a comment/suggestion here. Given there is already a standard for manuka honey (cf. lines 40 – 43), is there a need for a pollen-based standard? What value could this bring above the existing test? I think it would add to the impact of the paper if the authors could add a sentence or two which proposes why a standard including pollen morphology/melissopalynology could be beneficial for the industry. I’m aware that there are limitations to the existing MPI standard.

------We have modified the text to include a brief statement of the benefits of melissopalynology. The relationship of the MPI test to the actual manuka contribution to a honey is a complex and we suggest an as yet inadequately studied topic. We felt that to address this issue properly would be beyond the scope of our paper, in which we wished to report mainly basic pollen data which can be built upon in later analyses.

Line 94: I don’t really follow what is meant by “each taxonomic entity geographically apart”. Is it possible to make the meaning clearer here?

------text modified

Lines 176 – 180: I’m a little bit confused as to how this could happen if the images were manually selected, i.e. I would have assumed that images selected would have been those which were not only properly orientated, but also appeared to be of good quality, i.e., not fuzzy, no debris adhering to the pollen, no hollowing out by aggressive edge detection etc. I’m assuming that some fault in the image is what is meant by “imperfections”.

Perhaps a better/briefer way to capture this aspect of the methodology is to state that only properly oriented images and good quality images (i.e. images without artifacts, etc.) were selected for use.

Alternatively, if the preference is to maintain the text as is, then it would be helpful to know more about the nature of the imperfections in the image quality and why compactness factor was used to define what was retained/discarded.

------We have modified existing text to state that it was mainly poor focus images which caused the outlying results.

Line 308/333: a reference for the Coda package should probably be provided, i.e. Plummer et al (2006)

------Reference provided

Lines 444 – 457: Yes, I totally agree overall. Yes, the native feature set of the Classifynder is not optimised to distinguishing pairs or sets of very similar types. The DLCNN approach applied in the papers referenced in this section (48 & 49) is probably the most promising avenue, as these NNs essentially define their own features based what they determine to be the most discriminatory aspects of the classes, as provided in the training sets. This aspect might we worth adding to the discussion. Likewise, in [49], the images of manuka and kanuka were randomly oriented, rather than polar & equatorial, and yet the system was still able to differentiate them reasonably well. I know from experience that sifting through 1000s of images for correctly aligned ones is time consuming and would really limit the deployment of the Classifynder in the routine application of a manuka pollen standard. Therefore, the ability to use randomly oriented pollen images is probably necessary here, thus making DLCNNs the best option. Again, I suggest the authors consider rewording the discussion to make reference to these issues.

------This text has been modified to address the DLCNN advantages mentioned by the reviewer.

Reviewer #2: Dear Xun and co-authors,

You have done a thorough job in examining pollen of New Zealand Leptospermum and Kunzea. What I found though was your manuscript lost focus on what your aim was. My understanding of your manuscript is that you want to find pollen characters that can help discriminate between the two NZ Leptospermeae species so that you can identify if the honey is Manuka honey or not. 

------This was our primary purpose, but as a basic pollen morphology study it may have other and perhaps unforeseen applications in future. This is partly why we have included details of the individual Leptospermum morphotypes and Kunzea species.

I think you have done that in this paper and it has to do with pollen measurements. But that message gets lost with the deep discussion you make on the differences within each species/morphospecies. I think the differences that you identify within the species are nothing major too. I think that most of the inter-species information could be moved to supplemental data or removed completely. Focus on the differences between the genera.

------In aiming to test the difference between pollen of mānuka and kānuka, we had to cover the whole range of species and morphotypes, and other morphological features as well as size. If we did not check all of them, how could we conclude that the pollen characteristics of the two genera are persistent within each genus but different between them? This is the body of our research, which we have sought to report accurately. We have considered moving text to supplemental data, but think that this would not fairly represent the work, or be as useful to readers. However, we have attempted to reduce the length of text by removing some duplication or less relevant matter.

By focusing on the genera you also need to consider a couple of things.

In your introduction you say "in December 2017 the New Zealand Ministry for Primary Industries (MPI) chose a purity test for manuka honey based on assay of characteristic chemical marker compounds and DNA of manuka pollen". If this is the government policy for determining manuka honey, then why are you trying to identify pollen characters that would help identify manuka honey? Could you not say that DNA analysis of honey would unequivocally tell you if Leptospermum scoparium is in the honey? Why do palynology analyses? Could it be quicker and cheaper than a DNA test. If yes then say so.

------This study was commenced before MPI brought out their requirements for EXPORT manuka honey (note that there is no internal NZ standard, although there are NZ industry recommendations for combined Leptospermum-Kunzea pollen content dating back to 2008). We believe there remain advantages for pollen analysis and have now included a brief explanation in the introduction. See also our response to the editor.

This paper suggests that each genus probably has more than one NZ species. I'm not question that here. However, if there is more than one species in Leptospermum then does their pollen all produce manuka honey? If not then which ones. That would make identifying manuka honey using pollen even trickier.

------At the present time, no distinction between possibly different NZ Leptospermum species is made in identifying manuka honey. Such a proposition would probably not be supported by the NZ honey industry.

One last consideration is that this study only used native NZ plants from herbarium specimens. You can correct me if I am wrong, but aren't there now Australian species of Leptospermum naturalised in NZ? How would a real world example to identify manuka honey work if you also have Australian Leptospermum pollen muddying the honey. How would you unequivocally say that the naturalised Leptospermum pollen wasn't in the honey?

------We know of 7 casual or fully naturalised Leptospermum species (data from Schönberger et al. (2021) and Ogle et al. (2020), and personal observation of P. de Lange): 

L. laevigatum - locally common in northern Bay of Plenty, Matakana Island especially

L. minutifolium - Auckland only - collected once

L. petersonii - Auckland - common garden plant scarce in the wild

L. morrisonii - Auckland / Nelson (Golden Bay) - very uncommon

L. polygalifolium subsp. polygalifolium – locally common in urban Auckland

L. spectabile - Northland - collected once not seen again

L. variabile - Auckland (Western Springs) collected once and not seen again

None of these are sufficiently widespread to be likely to contribute much to the honey industry, although it is possible that hives on Matakana Island will collect some L. laevigatum honey. We have now noted this in the text. Study of the pollen of this species is an omission which we cannot now encompass.

If you can address these issues in the manuscript then I think it will be more impactful and of more use in honey pollen identification.

Below are comments made while reading the manuscript.

Regards,

Andrew Thornhill

Page 2 Line 14 - I think species needs to be changed to taxa to make grammatical sense. As it reads now it is "Myrtaceous species

Leptospermum scoparium and Kunzea spp (i.e. species)"

------Accepted and text changed.

Page 2 Line 19 - Is 'a' needed before Classifynder?

------Text corrected.

Page 2 Line 37 - 'scoparium' isn't italicised

------Text corrected.

Page 2 Line 38 - Technically this statement is true in that Kunzea is the closest relative to Leptospermum in NZ. I wouldn't say Kunzea is the sister genus of Leptospermum however because phylogenies show them to be separated by at least Asteromyrtus and Neofabricia. See the phylogenies of Maurin et al 2021 and Thornhill et al 2015.

------We have modified the text accordingly.

Page 5 line 87 - Do all morphotypes of Leptospermum scoparium form Manuka honey? If they don't, what is the point of looking for different types within Leptospermum scoparium?

------No distinction is at present made between the different possible L. scoparium morphotype sources for mānuka honey.

Page 12, line 242 - This paper's goal is to determine whether the plants that make manuka honey have different pollen to those that don't. The problem I see here is that if there are 10 species of Leptospermum in NZ - do all of these species form pollen that makes manuka honey. If they don't then which Leptospermum species/morphotypes do? Then the next question is do those Leptospermum species have unique pollen?

------As noted above, all NZ Leptospermum plants are at present assumed to be eligible sources. It is known that there may be differences in the chemistry of the resulting honeys (e.g., in the DHA content). In this paper, we demonstrate that there are very limited differences in the pollen morphology between the NZ Leptospermum populations. For NZ honey pollen analysis, this is an important finding of our paper, as also that pollen of all NZ Kunzea pollen can be distinguished from that of all NZ Leptospermum (at least on a reasonably applicable statistical basis).

Page 12, Line 248 - I would be a bit cautious in using pollen size to help define different species. It looks like the pollen size differences between your different morphospecies is 2um if that. If I was to gather the same pollen and measure I have a feeling my measurements would have a greater error margin than 2um. The ability to recreate your pollen measurements means that everyone would need to use the same magnification, the same eye lens and the same microscope. Given that your size differences between plants is so small I think it is hard to justify the differences.

------This paragraph has been removed to save space. The relationship of our pollen results (which include other morphological characters as well as size) to taxonomy will be better assessed when the taxonomy of NZ Leptospermum is further understood. 

Other workers seeking to research pollen morphology of these taxa would have to use a uniform technique, but reproduction of our optics would not be necessary as it is relative sizes and shapes that are important rather than precise measurements.

Page 11 Table 1. How can a feature be obviously seen & not seen at the same time? I'm confused by this table

------Some features are variable, evident in some collections or individual specimens but not so in others. We have scored both states in our table. An explanatory remark has been added to the caption.

Page 14, line 286 - This sentence highlights why you need to be cautious using pollen in taxonomy. If you don't use the exact same processing methods and microscopes then you are likely to get a different measurement. The difference in the same species between Pete's observations and the current one is around 3-4 um. This study shows that morphotypes have a difference of 1um or less. If a third study was done they would need to sample all morphotypes to come up with a range of smallest to largest. You wouldn't be able to sample just one morphotype and confidently conclude that you have X morphotype.

------We agree, consistent technique is undoubtedly important. The difference between Peter de Lange’s previous results and our new measurements is principally due to his pollen not being acetolysed. This increases variability due to varying degrees of hydration of the protoplasm, compared to acetolysed pollen where protoplasm is no longer present. We have now removed this paragraph to reduce the length of the paper.

Page 15 line 310 - If the goal of the paper is to show the difference between Leptospermum and Kunzea pollen, then I think it appears quite simple. If a pollen is 14.32-17.93 in equatorial diametre then it is Kunzea. If it is 19-22.18 then it is Leptospermum. Anything in between can not be confidently assigned to a genus. The same would apply for measurements in polar view. It's the extreme of each measurement where there is no overlap between the two genera that are the unique pollen feature.

------This may be true for a distinction based solely on pollen size, but separation of the two genera can be improved by including the differences in pollen shape and surface texture. This is evident from the DLCNN results as well as our visual observations.

Page 17 line 362. Extra comma before full stop at end of sentence 1.

------Corrected

Page 18 line 381-382. If they can be separated by 1 St. Dev., why would you use another measure that is less accurate?

------Text amended to remove the 2 std deviation statement.

Page 19 line 396 - I think this is the incorrect citation. It should be this one -

Thornhill, A. H., Wilson, P. G., Drudge, J., Barrett, M. D., Hope, G. S., Craven, L. A., et al. (2012). Pollen morphology of the Myrtaceae. Part 3: tribes Chamelaucieae, Leptospermeae and Lindsayomyrteae. Aust. J. Bot. 60, 225–259. doi:10.1071/BT11176.

------We have changed the citation as indicated.

Page 19 line 412. Were they able to be separated or not? If there's enough variation within each that they can't be separated, does that not suggest they are a conglomerate?

------We do not understand this remark. Many genera, like Leptospermum and Kunzea, display fairly uniform pollen morphology among well-delineated species. This does not mean that they are not distinct genetic entities, or that some intrageneric groupings may not still be recognised. As noted elsewhere in Discussion, Classifynder measurements may not reveal some of the distinctions possible using visual observation.

Page 20 line 424 Is there a range of variation that encompasses this? What is the level of distinction that you feel demonstrates the shift from intraspecific to interspecific variation?

------The text here is discussing the possible origins of intraspecific variation in size. It discounts freshness of herbarium material, and suggests that variation may result from varied pollen maturity. We cannot quantify this.

Page 20 line 436 Do your separated populations interbreed at all? Are there other Leptospermum spp. or closely related species?

------Kunzea species do interbreed, and their progeny is fully fertile, but hybrids are scarce outside sites of protracted human disturbance (de Lange et al. 2005, de Lange 2014). The situation with Leptospermum is at present uncertain. We have removed this paragraph, as on reconsideration we think that contaminant pollen from nearby genetically different but related populations will not be a numerically significant cause of variation in size measurements. 

Page 21 line 455. If Classyfinder has such a high accuracy, then it should have been able to detect any valid differences between the pollen morphologies in the dataset.

------We have expanded discussion of DLCNN following the remarks of Reviewer 1. However, note that Classifynder may have physical or algorithmic deficiencies, e.g., optical resolution of the 40x microscope objective used by Classifynder is less than that of the 100x oil immersion objective used by us for observation of surface texture. See remarks by Reviewer 1 for lines 444 et seq. In general, we have attempted to discover variation within NZ Leptospermum and Kunzea by understandable human observation rather than rely only on a “black box” algorithm. 

Page 21 line 459. Has this been demonstrated effectively? The paper seems to have lost focus on this purpose. The interest in distinguishing the Leptospermum scoparium populations is clear, but the abstract doesn't seem to accurately represent the paper's contents.

------We disagree with this assessment. However, we have attempted to improve the focus of the paper by deleting some less relevant text and by improving the introduction.

---

## [Decision Letter · Decision Letter 1]

20 May 2022

Discrimination of pollen of New Zealand mānuka (Leptospermum scoparium agg.) and kānuka (Kunzea spp.) (Myrtaceae)

PONE-D-21-39233R1

Dear Dr. Li,

We’re pleased to inform you that your manuscript has been judged scientifically suitable for publication and will be formally accepted for publication once it meets all outstanding technical requirements.

Kind regards,

Wolfgang Blenau

Academic Editor

PLOS ONE

Additional Editor Comments (optional):

Reviewers' comments:

Reviewer's Responses to Questions

**Comments to the Author**

1. If the authors have adequately addressed your comments raised in a previous round of review and you feel that this manuscript is now acceptable for publication, you may indicate that here to bypass the “Comments to the Author” section, enter your conflict of interest statement in the “Confidential to Editor” section, and submit your "Accept" recommendation.

Reviewer #1: All comments have been addressed

Reviewer #2: All comments have been addressed

2. Is the manuscript technically sound, and do the data support the conclusions?

Reviewer #1: (No Response)

Reviewer #2: Yes

3. Has the statistical analysis been performed appropriately and rigorously? 

Reviewer #1: (No Response)

Reviewer #2: Yes

4. Have the authors made all data underlying the findings in their manuscript fully available?

Reviewer #1: (No Response)

Reviewer #2: Yes

5. Is the manuscript presented in an intelligible fashion and written in standard English?

Reviewer #1: (No Response)

Reviewer #2: Yes

6. Review Comments to the Author

Reviewer #1: (No Response)

Reviewer #2: I think that you have greatly improved the paper by reducing the amount of discussion on the differences between morphospecies. I could only find one correction for you. It is

Line 418 - delete 'of''

7. PLOS authors have the option to publish the peer review history of their article (what does this mean?). If published, this will include your full peer review and any attached files.

Reviewer #1: **Yes: **Katherine Holt

Reviewer #2: **Yes: **Andrew Thornhill

---

## [Editor Report · Acceptance letter]

25 May 2022

PONE-D-21-39233R1 

Discrimination of pollen of New Zealand mānuka (*Leptospermum scoparium* agg.) and kānuka (*Kunzea* spp.) (Myrtaceae) 

Dear Dr. Li:

I'm pleased to inform you that your manuscript has been deemed suitable for publication in PLOS ONE. Congratulations! Your manuscript is now with our production department. 

Kind regards, 

on behalf of

Dr. Wolfgang Blenau 

Academic Editor

PLOS ONE